# Drought may exacerbate dryland soil inorganic carbon loss under warming climate conditions

Jinquan Li[1], Junmin Pei[1,2], Changming Fang[1], Bo Li [1,3] & Ming Nie [1] ✉

Low moisture conditions result in substantially more soil inorganic carbon (SIC) than soil organic carbon (SOC) in drylands. However, whether and how changes in moisture affect the temperature response of SIC in drylands are poorly understood. Here, we report that the temperature sensitivity of SIC dissolution increases but that of SOC decomposition decreases with increasing natural aridity from 30 dryland sites along a 4,500 km aridity gradient in northern China. To directly test the effects of moisture changes alone, a soil moisture control experiment also revealed opposite moisture effects on the temperature sensitivities of SIC and SOC. Moreover, we found that the temperature sensitivity of SIC was primarily regulated by pH and base cations, whereas that of SOC was mainly regulated by physicochemical protection along the aridity gradient. Given the overall increases in aridity in a warming world, our findings highlight that drought may exacerbate dryland soil carbon loss from SIC under warming.

Drylands are regions where the aridity index is below 0.65 (ref. 1), and they cover approximately 41% of the terrestrial surface[2]. Global warming is predicted to increase the aridity of terrestrial ecosystems worldwide[3], resulting in an 11–23% increase in the global area of drylands within the 21st century[4]. However, these ecosystems are considered fragile[5] and vulnerable to aridity changes[6,7]. The presence of soil inorganic carbon (SIC) is primarily regulated by parent material[8], and the low moisture conditions in drylands favor a higher ratio of SIC to soil organic carbon (SOC) given that plant inputs are low, with approximately 2–10 times more SIC storage than SOC in these ecosystems[9]. Global SIC is estimated to be approximately 1237 Pg C to a depth of 2 m (ref. 10), with as much as 95% of the SIC stored in drylands[11]. Historically, SIC is generally considered very stable[12], and thus, previous studies on the temperature response of soil carbon (C) have focused entirely on SOC[13,14]. Similar to the biochemical reaction of SOC, increasing evidence has indicated that the SIC process is temperature-dependent[15], and SIC, therefore may contribute substantially to warming-induced soil C losses[8,16,17]. A recent synthesis of

28 studies has shown that SIC-derived $CO_2$ contributed 27% of total $CO_2$ emissions from calcareous soils[16]. However, it is not yet clear how temperature changes affect this vast SIC pool in drylands, where more frequent and extreme heat waves are predicted during the 21st century[7].

The general projected trend of global drylands is toward drying under climate change[18,19]. Although it is well known that drying decreases soil $CO_2$ emissions by inhibiting both SOC decomposition and SIC dissolution[16], how changes in moisture affect the temperature response of these two processes remains unexplored. Previous studies on soil moisture effects on the temperature sensitivity ($Q_{10}$) of total $CO_2$ emissions ($Q_{10\_total}$) have revealed inconsistent results; positive[20], negative[21], and no effect[22] have all been reported. The causes of this controversy may partly stem from the confounding moisture effects on the $Q_{10}$ of SOC-derived $CO_2$ emissions ($Q_{10\_SOC}$) and that of SIC-derived $CO_2$ emissions ($Q_{10\_SIC}$)[16]. A decrease in soil moisture may suppress $Q_{10\_SOC}$ by decreasing substrate availability[23,24]. Soil moisture can also exert crucial controls on $Q_{10\_SOC}$ by regulating

[1]Ministry of Education Key Laboratory for Biodiversity Science and Ecological Engineering, National Observations and Research Station for Wetland Ecosystems of the Yangtze Estuary, School of Life Sciences, Fudan University, Shanghai 200438, China. [2]College of Life Sciences, Shanghai Normal University, Shanghai 200234, China. [3]Ministry of Education Key Laboratory for Transboundary Ecosecurity of Southwest China, School of Ecology and Environmental Science, Yunnan University, Kunming 650504 Yunnan, China. ✉e-mail: mnie@fudan.edu.cn

physicochemical protection[25] and microbial communities[26]. In comparison, the release of C during SIC dissolution is mainly a chemical process, which is described by the following equations:

$$CaCO_3 + H^+ \leftrightarrow Ca^{2+} + HCO_3^-, \tag{1}$$

$$HCO_3^- + H^+ \leftrightarrow CO_2 + H_2O. \tag{2}$$

Soil moisture changes can directly drive the $CaCO_3$–$CO_2$–$HCO_3^-$ equilibrium equations to promote or inhibit $CaCO_3$ dissolution[17,27], indicating that moisture effects on SIC dissolution are not linear or stagnant and that reprecipitation processes may also occur. For example, a decrease in soil moisture would inhibit $CaCO_3$ dissolution[16], and the observed net SIC-derived $CO_2$ emissions can be especially low under low moisture conditions, as $CO_2$ is also consumed during carbonate dissolution[28]. Soil moisture can also indirectly affect SIC dissolution by mediating soil pH and/or base cations (e.g., $Ca^{2+}$ and $Mg^{2+}$)[16] that can shift the reactions represented in Eqs. (1) and (2). Accordingly, a drop in soil $H^+$ owing to the enhancement of soil pH resulting from soil moisture decrease[29] will lead to the reactions represented in Eqs. (1) and (2) to proceed to the left, leading to low SIC-derived $CO_2$ emission rates. Given that a low $CO_2$ emission rate is more sensitive to environmental changes (e.g., temperature), moisture decrease and/or pH or base cation increase may enhance the temperature response of SIC dissolution.

Moreover, as SOC decomposition affects the soil $CO_2$ concentration[30], factors (e.g., physicochemical protection[25] and substrate[24]) that affect SOC decomposition may also mediate SIC dissolution and its temperature response. An increase in SOC decomposition will lead to more active $CO_2$ sources in soil for $HCO_3^-$ production, which will restrict $CaCO_3$ dissolution. Consequently, soil physicochemical protection and substrate can regulate SIC processes by mediating SOC decomposition. Although mineral protection of Ca bridges and/or Fe oxides has been shown to largely inhibit SOC decomposition and its temperature sensitivity[14,31], its effects on $Q_{10\_SIC}$ remain unknown. These direct and indirect moisture effects are not independent but coexist temporally and spatially[16], while the main drivers and their differences in regulating $Q_{10\_SOC}$ and $Q_{10\_SIC}$ are poorly understood. Until now, no attempt has been made to examine moisture effects on $Q_{10\_SOC}$ and $Q_{10\_SIC}$ across a large moisture gradient in drylands. This knowledge gap urgently needs to be filled since future climate change is predicted to largely affect dryland moisture regimes[18] and, consequently, the climate–C cycle feedbacks in these water-limited ecosystems.

Here, we hypothesized that (i) $Q_{10\_SOC}$ decreased, but $Q_{10\_SIC}$ increased with decreasing moisture content, and (ii) $Q_{10\_SOC}$ was mainly regulated by physicochemical protection, while $Q_{10\_SIC}$ was primarily regulated by chemical properties (e.g., pH and cation exchange capacity (CEC)). To test these hypotheses, we conducted two experiments (Fig. 1): a natural aridity gradient and a moisture control treatment. In the first experiment, soil moisture regime differences were evaluated by sampling soils from 30 sites across a wide aridity index (the ratio of precipitation to potential evapotranspiration, ranging from 0.04 to 0.59) along an approximately 4500 km east–west transect in the drylands of northern China; in this experiment, $Q_{10\_SOC}$ and $Q_{10\_SIC}$ were determined with field moisture conditions. Considerable differences in $Q_{10}$ and its controls were expected to exist throughout the soil profile[32,33] owing to the large differences in soil biotic and abiotic factors[34,35]. To test whether moisture effects on $Q_{10\_SOC}$ and $Q_{10\_SIC}$ persist among different soil depths, soils from the topsoil (0–10 cm) and subsoil (35–50 cm) were collected at each site. In the second experiment, to directly test the effects of only moisture changes on $Q_{10\_SOC}$ and $Q_{10\_SIC}$, we conducted a moisture control experiment by incubating soils under different moisture conditions of

20%, 40%, and 60% water holding capacity (WHC). To determine the main drivers and their differences associated with variations in $Q_{10\_SOC}$ and $Q_{10\_SIC}$ that were determined under field moisture conditions along the aridity gradient, we analyzed various potential factors related to climate (mean annual temperature (MAT) and aridity index), physical (SOC stored in particulate organic matter (OC-POM) and mineral-associated organic matter (OC-MAOM) fractions, and SOC associated with Ca bridges (OC-Ca) and Fe oxides (OC-Fe)), chemical (pH, CEC, $Ca^{2+}$, and $Mg^{2+}$) and substrate (quantity, quality and availability) properties.

## Results

### Moisture effects on SOC- and SIC-derived $CO_2$ emissions and their $Q_{10}$ values

Isotopic measurement of $^{13}C$ content is considered an effective approach for distinguishing different $CO_2$ sources[16]; SOC is less enriched with heavier $^{13}C$ than SIC, and thus, the $\delta^{13}C$ value of SOC-derived $CO_2$ differs substantially from that of SIC-derived $CO_2$[15,17]. In this study, natural isotope technology was adopted to separate total soil $CO_2$ emissions into SOC- and SIC-derived sources. Across the aridity gradient in drylands, the average contribution of SIC-derived $CO_2$ to total $CO_2$ emissions was 7.2% and 11.1% in the topsoil and subsoil, respectively, at 20 °C (Supplementary Fig. 1). SOC- and SIC-derived $CO_2$ emissions differed as a function of soil moisture, showing that they both decreased with increasing aridity (Supplementary Fig. 2). Similar results were observed from the moisture control experiment, with lower emission rates under lower moisture contents (Supplementary Fig. 3).

To reveal moisture effects on the temperature response of SOC and SIC, we first evaluated changes in $Q_{10\_SOC}$ and $Q_{10\_SIC}$ along the aridity gradient; to do this, soils were incubated under field moisture conditions. Opposing patterns of $Q_{10\_SOC}$ and $Q_{10\_SIC}$ were observed in response to aridity changes, showing that in both soil layers, $Q_{10\_SOC}$ decreased significantly with drying (that is, decreasing aridity index), whereas $Q_{10\_SIC}$ increased significantly with drying ($P < 0.001$, Fig. 2). This was further verified by the moisture control experiment; given the predicted overall aridity increase in drylands in a warmer world, soils were incubated under different moisture contents of 20%, 40%, and 60% WHC. The moisture control experiment also showed that $Q_{10\_SOC}$ was lower under lower experimental moisture conditions, but the opposite was true for $Q_{10\_SIC}$ ($P < 0.01$, Fig. 3).

### Factors regulating the $Q_{10\_SOC}$ and $Q_{10\_SIC}$ along the aridity gradient

We next explored factors that regulate the variations in the $Q_{10}$ of SOC- and SIC-derived $CO_2$ emissions along the aridity gradient. Potential influencing factors were determined, comprising groups of factors related to climate (i.e., MAT and aridity index), physical (i.e., SOC stored in the POM and MAOM fractions and SOC associated with Ca oxides and Fe bridges), chemical (i.e., pH, CEC, $Ca^{2+}$ and $Mg^{2+}$), substrate (i.e., substrate quantity of SOC and SIC contents and substrate availability of C availability index (CAI), and substrate quality of SOC decomposability ($D_{SOC}$)). Correlation analysis revealed that $Q_{10\_SOC}$ was linked to climate, soil physical, and substrate at both soil depths (Fig. 4). Specifically, $Q_{10\_SOC}$ was positively correlated with the aridity index, OC-POM, and CAI but negatively correlated with OC-MAOM, OC-Fe, OC-Ca, and $D_{SOC}$ (Fig. 4). $Q_{10\_SIC}$ was linked to climate and soil chemical properties at both soil depths, showing that $Q_{10\_SIC}$ was negatively correlated with the aridity index but positively correlated with pH, $Ca^{2+}$, and $Mg^{2+}$ (Fig. 4). A similar phenomenon was observed when $Q_{10\_SOC}$ and $Q_{10\_SIC}$ were determined under common moisture conditions (Supplementary Fig. 4).

A structural equation modeling (SEM) was further constructed to assess the direct and indirect effects of these factors (i.e., climate, physical, chemical, and substrate properties) on the $Q_{10}$ of SOC- and

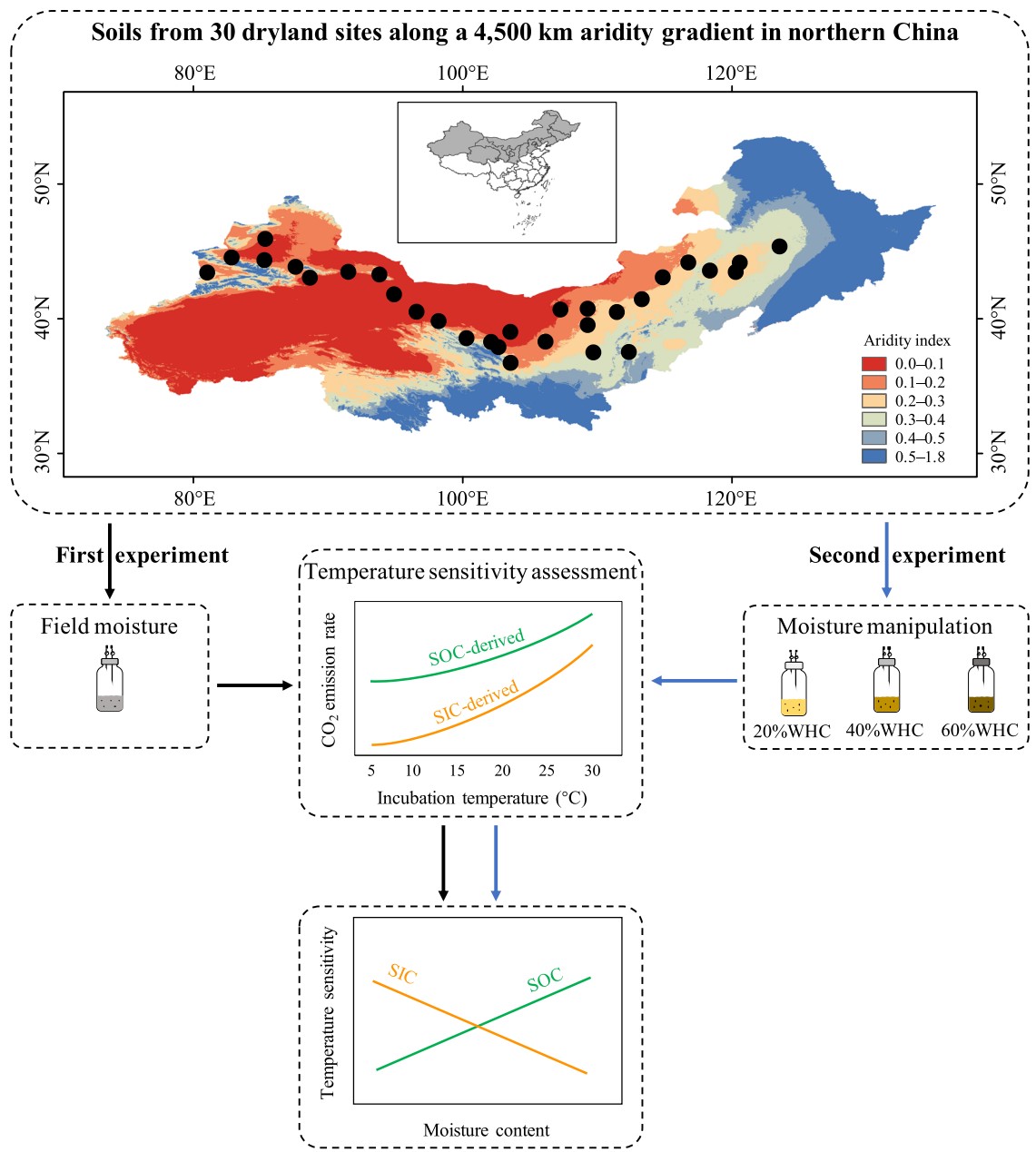

**Fig. 1 | Flow chart to test how changes in moisture affect the temperature sensitivity ($Q_{10}$) of SOC- and SIC-derived $CO_2$ emissions in drylands.** Soils were collected from 30 dryland sites along a 4500 km aridity gradient in northern China; the background map of China, made with the National Geomatics Center of China (https://www.ngcc.cn/ngcc/), is publicly available. For the first experiment, differences in the soil moisture regime were evaluated along the aridity gradient, and $Q_{10}$ was determined under field moisture conditions. For the second experiment, a moisture control experiment was conducted, and $Q_{10}$ was determined under 20%, 40%, and 60% water holding capacity (WHC). We hypothesized that the $Q_{10}$ of SOC- and SIC-derived $CO_2$ emissions would respond differently to moisture changes.

SIC-derived $CO_2$ (Fig. 5). The results showed that among all the groups of factors tested, climate, soil physical and substrate had direct effects on $Q_{10\_SOC}$ (Fig. 5a), with greater total effects of climate and soil physical variables than other factors (Fig. 5c). For $Q_{10\_SIC}$, climate and soil chemical properties had significant direct effects ($P < 0.05$, Fig. 5b), and they had greater total effects than other factors (Fig. 5d). Similar results were observed in the subsoil (Supplementary Fig. 5).

## Discussion

On the basis of large-scale sampling and isotope approaches, over the short-term time scale, we observed substantial SIC contributions to soil total $CO_2$ emissions. In long-term monitoring studies at national or even global scales, some recent studies have also observed SIC losses[36,37]. Given that SIC accumulation usually takes substantially more time than SOC[38], SIC losses are thus more impactful for the soil C reservoirs than that of SOC in these water-limited ecosystems. Moreover, we observed that absolute SOC- and SIC-derived $CO_2$ emissions both decreased, but the contribution of SIC to total $CO_2$ emissions increased with decreasing moisture along the aridity gradient; this might be because the microbial process of SOC decomposition is more sensitive to moisture changes than the chemical process of SIC dissolution[17,39]. Our results thus indicate that the dissolution of SIC is more important than previously thought in regulating atmospheric $CO_2$ concentrations[8], and if future climate change accelerates aridity in drylands[18], the contribution of SIC-derived $CO_2$ to total $CO_2$ emissions may become even more substantial.

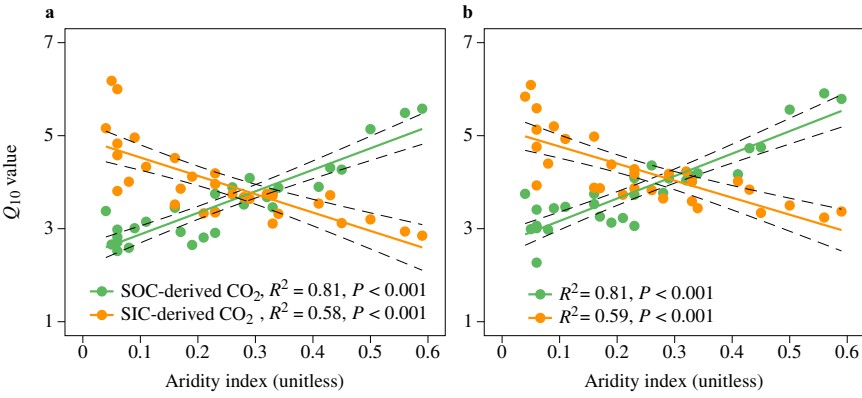

**Fig. 2 | Different responses of the temperature sensitivity ($Q_{10}$) of SOC- and SIC-derived $CO_2$ emissions to aridity changes.** Relationships of the $Q_{10}$ of SOC- and SIC-derived $CO_2$ emissions with aridity in the topsoil (0–10 cm, **a**) and subsoil (35–50 cm, **b**). Linear regression was used, and the dashed lines surrounding the regression lines correspond to the 95% confidence interval of the correlation. $Q_{10}$ was estimated under field moisture conditions.

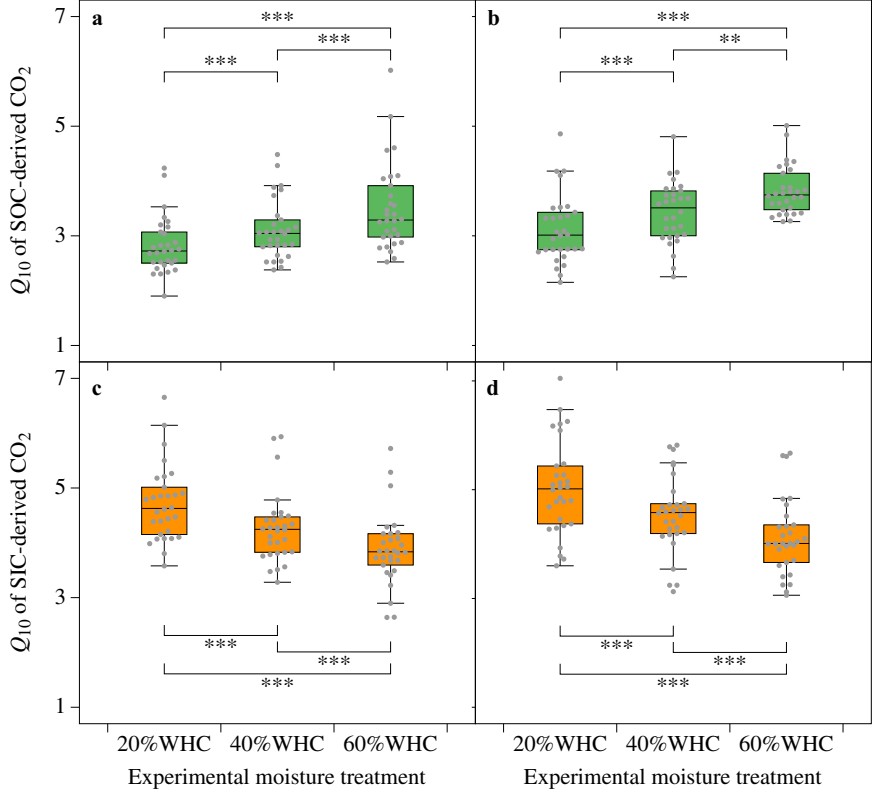

**Fig. 3 | Differences in the temperature sensitivity ($Q_{10}$) of SOC- and SIC-derived $CO_2$ emissions among different experimental moisture treatments.** **a**, **b** Differences in the $Q_{10}$ of SOC-derived $CO_2$ in the topsoil (0–10 cm, **a**) and subsoil (35–50 cm, **b**) among different moisture conditions. **c**, **d** Differences in the $Q_{10}$ of SIC-derived $CO_2$ in the topsoil (**c**) and subsoil (**d**) among different moisture conditions. The horizontal lines inside the box represent the median, the ends of the boxes represent the first and third quartiles, and the whiskers show the interquartile range from the first and third quartiles. The gray dots indicate values for each of the 30 sites. All statistics were derived from $n = 30$ independent samples, and statistical significance was tested using a two-sided, paired-sample $t$ test. **, $P < 0.01$; ***, $P < 0.001$; WHC water holding capacity.

Temperature sensitivity represents a key parameter in many previously developed biogeochemical models that have simulated C emissions[40], and small inaccuracies in this parameter can result in large errors[41]. However, total $CO_2$ emissions were typically measured in the majority of previous studies on $Q_{10}$, and the calculated value was considered the $Q_{10}$ of SOC decomposition[13,14,32,42]. For acidic soils, measurements of total $CO_2$ emissions are sufficient for evaluating the $Q_{10}$ of SOC decomposition[33]; for calcareous soils, however, this would bias our understanding, as $Q_{10\_total}$ was higher than $Q_{10\_SOC}$. Although the absolute rate of SIC-derived $CO_2$ is low compared to that of SOC-

derived $CO_2$ emissions, the high-temperature sensitivity of SIC and the vast SIC stock in drylands[11] can also harbor great potential for regulating climate–C cycle feedbacks in drylands. Our results provide further evidence that moisture has opposite effects on the temperature response of SOC decomposition and SIC dissolution, which is a crucial step forward in gaining an understanding of climate–C cycle feedbacks in drylands. The general projected global trend for drylands predicts drying[18], and our findings of positive $Q_{10\_SOC}$–moisture but negative $Q_{10\_SIC}$–moisture relationships suggest that drought may exacerbate warming-induced soil C loss from inorganic C in drylands.

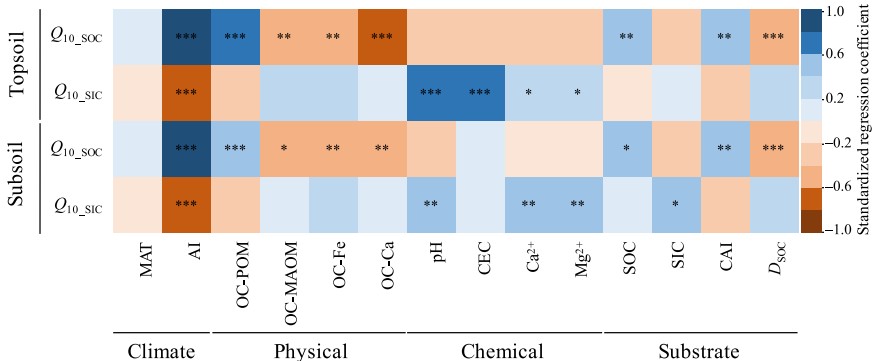

**Fig. 4 | Pearson correlations (*r*) of the temperature sensitivity (*Q*₁₀) of SOC- and SIC-derived CO₂ emissions with factors related to climate, physical, chemical, and substrate.** Corrected significance at $P < 0.001$ is represented with ***, $P < 0.01$ is represented with **, and $P < 0.05$ is represented with *. $Q_{10}$ was estimated under field moisture conditions. $Q_{10\_SOC}$ and $Q_{10\_SIC}$ $Q_{10}$ of SOC-derived and SIC-derived CO₂ emissions, respectively, MAT mean annual temperature, AI aridity index, OC-Ca and OC-Fe the contents of SOC, associated with Ca oxides and Fe bridges, respectively, OC-POM and OC-MAOM the content of SOC stored in the POM and MAOM fraction, respectively, CEC cation exchange capacity, CAI carbon availability index, $D_{SOC}$ SOC decomposability.

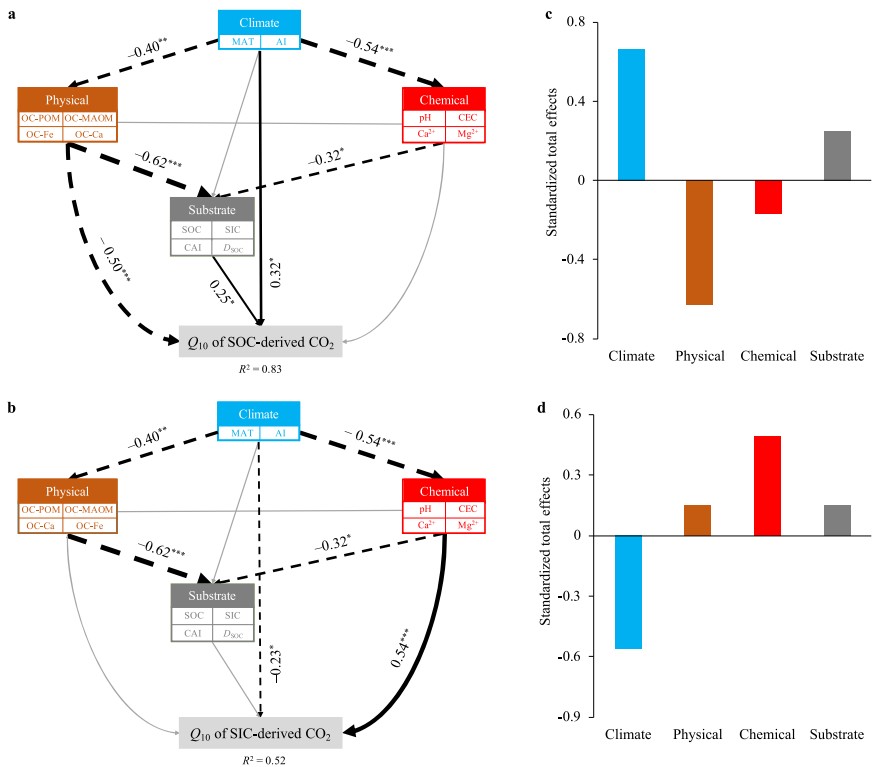

**Fig. 5 | Direct and indirect effects of climate, physical, chemical, and substrate properties on the temperature sensitivity (*Q*₁₀) of SOC- and SIC-derived CO₂ in the topsoil (0–10 cm). a, b** Structural equation modeling (SEM) was conducted for the $Q_{10}$ of SOC-derived CO₂ (**a**) and SIC-derived CO₂ (**b**). $Q_{10}$ was estimated under field moisture conditions. Black dotted and solid arrows indicate negative and positive relationships, respectively, and gray arrows indicate non-significant relationships; the arrow width represents the strength of the relationship, with the adjacent numbers representing the standardized path coefficients. The multiple-layer rectangles indicate the first component from the principal component analyses conducted for the climate, physical, chemical, and substrate properties. **c, d** The standardized total effects of different factors on $Q_{10}$ of SOC-derived CO₂ (**c**) and SIC-derived CO₂ (**d**) derived from the SEM. MAT mean annual temperature, AI aridity index, OC-Ca and OC-Fe the contents of SOC associated with Ca oxides and Fe bridges, respectively, OC-POM and OC-MAOM the content of SOC stored in the POM and MAOM fraction, respectively, CEC cation exchange capacity, CAI carbon availability index, $D_{SOC}$ SOC decomposability.

Total SOC- and SIC-derived CO₂ emissions are roughly estimated at 20.4 Pg C year⁻¹ in drylands (assuming that soil respiration in drylands accounts for 38.6% of global soil respiration[43], 60% of which is from the heterotrophic component[44]), and SIC-derived CO₂ contributes to approximately 27.0% of total CO₂ emissions[16]. Using $Q_{10\_SOC}$ to represent $Q_{10\_SIC}$ would underestimate warming-induced SIC-derived CO₂ emissions (3.2 Pg C year⁻¹) by approximately 25.6% compared to that estimated (4.3 Pg C year⁻¹) using the higher $Q_{10\_SIC}$ under 4 °C of warming. Moreover, considering the different responses of $Q_{10}$ to moisture changes ($Q_{10\_SOC}$ decreases by 0.47 and $Q_{10\_SIC}$ increases by 0.39 per 0.1 decrease in the aridity index; Fig. 2), the net increase in soil C due to the lower $Q_{10\_SOC}$ (1.5 Pg C year⁻¹) would be offset by approximately 26.7% due to the higher $Q_{10\_SIC}$ (0.4 Pg C year⁻¹) under 0.1 decreases in the aridity index. Consequently, although these are rough

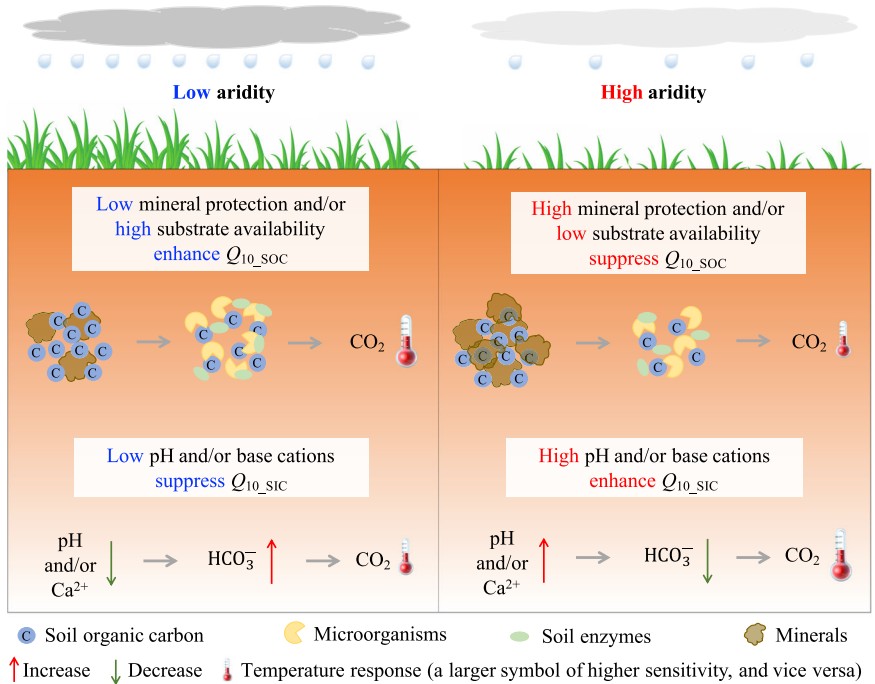

**Fig. 6 | Conceptual diagram showing the differential responses and controls of the temperature sensitivity ($Q_{10}$) of SOC and SIC to aridity changes in dryland ecosystems.** SOC decomposition temperature sensitivity ($Q_{10\_SOC}$) decreases with increasing aridity; this is mainly attributed to the increases in mineral protection and/or decreases in substrate availability. However, SIC dissolution temperature sensitivity ($Q_{10\_SIC}$) increases with increasing aridity, which is mainly attributed to the increases in soil pH and/or base cations.

estimates, they highlight the importance of separately representing $Q_{10\_SOC}$ and $Q_{10\_SIC}$ and their different responses to moisture changes to improve projections of climate−C cycle feedback in drylands.

This study has further identified differential mechanisms of $Q_{10\_SOC}$ and $Q_{10\_SIC}$ along the aridity gradient. Soil physicochemical protection primarily regulated the temperature sensitivity of SOC decomposition. The effect of soil physicochemical protection on $Q_{10\_SOC}$ could be due to the constraints of mineral protection on SOC availability and/or enzyme activity. Specifically, MAOM fractions can restrict oxygen diffusion and lead to the compartmentalization of organic C substrates from enzymes, and these processes can be enhanced by Ca bridges and/or Fe oxides[31]. In addition, Ca bridges and/or Fe oxides can constrain substrate availability by forming inner- and outer-sphere cation bridging between the negatively charged phyllosilicates and SOC[45]. Either of these processes may suppress the temperature response of SOC decomposition[24,34]. Consistent with this speculation, $Q_{10\_SOC}$ was positively correlated with OC-POM but negatively associated with OC-MAOM, OC-Fe, and OC-Ca along the aridity gradient (Fig. 4). In addition to physicochemical protection, aridity-induced changes in substrate also exerted roles in regulating $Q_{10\_SOC}$. Consistent with previous studies on moist soils[46,47], we observed that low substrate quality was associated with high $Q_{10\_SOC}$, as indicated by the negative relationship of $Q_{10\_SOC}$ with $D_{SOC}$ (Fig. 4), suggesting that the C quality-temperature hypothesis[13] is also applicable in these water-limited ecosystems.

However, $Q_{10\_SIC}$ was primarily regulated by aridity-induced changes in the soil chemical properties of soil pH and base cations along the aridity gradient, showing that higher pH and CEC enhance the temperature response of SIC. This is because a higher pH and/or base cation (e.g., $Ca^{2+}$ and $Mg^{2+}$) concentration may enhance the reverse reactions represented in Eqs. (1) and (2) toward the absorption of $CO_2$ into the soil solution. Given that a low $CO_2$ emission rate might be more sensitive to temperature changes, high pH and/or base cations can thus enhance $Q_{10\_SIC}$. Soil pH largely determines the stability of SIC[48], and we observed a positive correlation between pH and

$Q_{10\_SIC}$ but not $Q_{10\_SOC}$; this demonstrates that the usually observed positive pH−$Q_{10}$ relationship that measures total $CO_2$ emissions[33,49] may result from pH-induced changes to the temperature responses of SIC dissolution but not to microbial processes of SOC decomposition. The generally predicted drying may increase soil pH in drylands and thus further enhance the temperature response of SIC. Although an overall increase in aridity is predicted in drylands in a warmer world[18], some dryland regions are becoming increasingly prone to flooding[50], leading to increases in pedogenic carbonate accumulation[51]; however, flooding may also result in losses of both SOC and SIC through soil erosion in these areas[52].

In conclusion, our findings provide evidence for opposing moisture effects on the temperature response of SOC and SIC in drylands, suggesting that drying will further enhance the temperature response of SIC but weaken that of SOC (Fig. 6). This may partly explain the substantial loss of SIC pools over the past few decades on the continental scale[53] under conditions of drying. Additionally, we identified differential mechanisms regulating the temperature responses of SOC and SIC (Fig. 6). As drought is expected to enhance soil alkalinity[54], warming-induced SIC losses may be gradually enhanced by ongoing drought. In contrast, global changes in widespread nitrogen (N) deposition and/or acid rain are expected to promote soil acidification[55], possibly weakening warming-induced SIC losses. Although some studies have shown that soil C in drylands is resistant to N deposition[56,57], a recent study concluded that global N fertilization results in releases of $7.5 \times 10^{12}$ g C year$^{-1}$ from carbonates[37], and this value may be even underestimated[8]. Nevertheless, our finding of the positive pH−$Q_{10\_SIC}$ relationship suggests that SIC losses attributed to N deposition or acid rain may be weakened in a drying world. Therefore, to gain a better understanding of climate−C cycle feedbacks at an ecosystem level in drylands, future work should assess the potential effects of multiple global change factors on soil physicochemical (e.g., physical protection and pH) and biological (e.g., microbial community composition and functions) conditions and consequently their linkages with soil organic and inorganic C cycling.

## Methods

### Study area and field sampling

Soils were collected from 30 sites along an approximately 4500 km east–west transect with a longitudinal gradient of 81.02–123.53°E in northern China (Fig. 1). The climate is predominantly arid and semiarid continental; this transect covers the majority of the drylands in China and is the main reservoir of SIC in China[38]. The MAT and MAP ranged from −1.2 to 10.0 °C and from 46 to 486 mm, respectively, resulting in a wide range of aridity, with aridity indices (the ratio of precipitation to potential evapotranspiration) ranging from 0.04 to 0.59 at these sites; no significant relationship between MAT and aridity index was observed along the aridity gradient ($P = 0.128$, Supplementary Fig. 6). Soil properties for these sites were also greatly different; for example, soil pH and SIC content were significantly and negatively correlated with aridity index ($P < 0.05$, Supplementary Fig. 7). Therefore, this transect provided an ideal natural platform for studying moisture effects on the temperature sensitivity of SOC and SIC in drylands.

For soil sampling, three random locations (>20 m apart from each other) were chosen at each site. Because of the differences in environmental constraints, soil physical and chemical properties, and substrate and microbial properties among soil depths[34,35], there were likely to be considerable differences in $Q_{10}$ and its controlling factors throughout the soil profile[32]. Thus, soils from different depths of the topsoil (0–10 cm) and subsoil (35–50 cm) were collected. Soil samples were then passed through a 2-mm sieve, and soils from the three random locations were gently mixed to produce a homogeneous composite sample for each depth at each site. The composite sample was divided into three subsamples: one part was air-dried and processed for measurements of soil physical, chemical, and some substrate properties; another part was stored at 4 °C for soil incubations; the third part was stored at −20 °C for some microbial property analyses.

### Temperature sensitivity assessments

The $Q_{10}$ of SOC- and SIC-derived $CO_2$ emissions for the 60 soils (30 sites × 2 soil depths) was determined using a laboratory incubation experiment with a short-term dynamic temperature ramping method[58], which could minimize the different depletion of soil C pools[59,60] and microbial adaptation[61] to different temperatures relative to separate soil incubations at different but constant temperatures. To test moisture effects on the $Q_{10\_SOC}$ and $Q_{10\_SIC}$, we used two experiments: 1) the soils were incubated under field moisture conditions, and the relationship of $Q_{10}$ with the aridity index was tested; and 2) the soils were incubated under some common moisture content conditions of 20%, 40%, and 60% WHC, and the difference in $Q_{10}$ among different moisture conditions was tested. For the first experiment, the soils were quickly sieved to 2 mm at 4 °C after they were transported to the laboratory to minimize moisture changes and were then incubated to estimate $Q_{10}$. For the second experiment, soils sieved to 2 mm were adjusted to different moisture conditions of 20%, 40%, and 60% WHC by adding deionized water (with a pH of approximately 7.3) and were then incubated. For soil incubation for both experiments, 50 g of dry-weight fresh soil, with four experimental replicates, was maintained under field moisture conditions for the first experiment or adjusted to different moisture conditions (20%, 40%, and 60% WHC) for the second experiment and incubated in 250 ml jars. After a 2-week pre-incubation period at 20 °C to minimize disturbances from soil packing and rewatering, the jars with soils were incubated at 5–30 °C with a stepwise increase of 5 °C to perform the dynamic temperature ramping incubation[32]. After the soils were adjusted to the new target temperature and equilibrated for 3 h, the soil containers were sealed and flushed with $CO_2$-free air following previous studies[39,62]; 15 ml of headspace gas was removed for the initial $CO_2$ analysis, and 15 ml of $CO_2$-free air was immediately injected into the jars to allow them to equalize to atmospheric pressure. After incubation for 4–72 h

(depending on the target temperature), another 15 ml gas sample was collected. The $CO_2$ concentration and $\delta^{13}C$ value of these gas samples were analyzed using a gas isotope analyzer (G2201-20i, Picarro, USA).

SOC- and SIC-derived $CO_2$ was determined using a two-end-member mixing model:

$$\delta^{13}C_{CO2\_total} = (1 - f_{SIC}) \times \delta^{13}C_{CO2\_SOC} + f_{SIC} \times \delta^{13}C_{CO2\_SIC} \quad (3)$$

where $f_{SIC}$ is the contribution of SIC-derived $CO_2$ to total $CO_2$ emissions and $\delta^{13}C_{CO2\_total}$, $\delta^{13}C_{CO2\_SOC}$ and $\delta^{13}C_{CO2\_SIC}$ are the $\delta^{13}C$ values for total $CO_2$ release, SOC-derived $CO_2$ and SIC-derived $CO_2$, respectively. We assumed that the $\delta^{13}C$ value was the same for SIC and SIC-derived $CO_2$ and for SOC and SOC-derived $CO_2$[15–17]. The $\delta^{13}C$ value of $CO_2$ was then corrected because of differences in the fractionation of $\delta^{13}C$ at different temperatures[63]. Moreover, the $\delta^{13}C$ value of $CO_2$ released was 4.4‰ lower than that of $CO_2$ in the soil because of the fractionation induced by molecular diffusion[64].

A prior test showed that SOC- and SIC-derived $CO_2$ emissions increased exponentially with increases in incubation temperature (the fitting coefficient $R^2 > 0.95$ for SOC-derived $CO_2$ emissions and $R^2 > 0.85$ for SIC-derived $CO_2$ emissions). The temperature sensitivity of SOC- and SIC-derived $CO_2$ emissions was then calculated as follows:

$$R = Be^{kT} \quad (4)$$

$$Q_{10} = e^{10k} \quad (5)$$

where $R$ is the rate of SOC-derived and SIC-derived $CO_2$ (µg C g$^{-1}$ soil h$^{-1}$), $T$ is the incubation temperature (°C), and $B$ and $k$ are model fitting parameters.

Although the soil incubation experiment allowed us to reveal a general pattern and the mechanisms of moisture effects on $Q_{10\_SIC}$ and $Q_{10\_SOC}$ across large scales, we were aware that there were some possible influences, such as $CO_2$-free air flushing and soil sieving. To test the possible effects of $CO_2$-free air flushing on $Q_{10\_SOC}$ and $Q_{10\_SIC}$ estimation, we conducted a supplementary experiment using six representative soils (see Supplementary Text). The results showed that $CO_2$-free air flushing significantly underestimated SOC- and SIC-derived $CO_2$ emissions ($P < 0.05$, Supplementary Fig. 8); this might be because some of the SOC- and SIC-derived $CO_2$ might go into the pore space of soil, leading to an underestimation of SOC- and SIC-derived $CO_2$ emissions. However, $CO_2$-free air flushing had no significant effects on $Q_{10\_SIC}$ and $Q_{10\_SOC}$ ($P > 0.05$, Supplementary Fig. 8); this might be because $Q_{10}$ is a ratio for the temperature response of $CO_2$ emissions, resulting in limited effects of $CO_2$-free air flushing on $Q_{10}$ values. Moreover, we conducted another supplementary experiment using intact and sieved soils to test the effects of soil sieving on $Q_{10\_SOC}$ and $Q_{10\_SIC}$ (see Supplementary Text). The results showed that sieving did not exert significant effects on SOC- and SIC-derived $CO_2$ emissions and their $Q_{10}$ values ($P > 0.05$, Supplementary Fig. 9). This might be because the soils in drylands are usually sandy, resulting in limited effects from sieving. A recent study has also shown that soil sieving had no substantial effects on the $Q_{10}$ of total $CO_2$ emissions[65].

### Climate variables

MAT and MAP data were obtained from the WorldClim (https://www.worldclim.org/). Aridity indices were obtained from the Global Aridity Index and Potential Evapotranspiration Climate database (https://cgiarcsi.community/).

### Soil property analyses

To explore the factors regulating the temperature response of SIC- and SOC-derived $CO_2$ along the aridity gradient, soil physical, chemical,

and substrate properties were determined for the 60 soil samples (30 sites × 2 soil depths) collected across the natural aridity gradient.

**Soil physical properties.** The physical properties of POM and MAOM and the SOC associated with Ca bridges (OC-Ca) and Fe oxides (OC-Fe) were determined. A fractionation technique was adopted to estimate the SOC stored in the POM and MAOM fractions. Air-dried soil was separated into light and high-density fractions with sodium polytungstate solution ($1.60\,g\,cm^{-3}$)[66]; the high-density fractions were then wet sieved to collect POM (>53 μm) and MAOM (<53 μm)[67]. Moreover, to determine OC-Ca and OC-Fe contents[68], the high-density fractions were extracted using $0.5\,M\,Na_2SO_4$ to release OC-Ca; the remaining residues were then extracted with citrate–bicarbonate–dithionite and sodium chloride for the treatment and control groups, respectively, and the differences in SOC content between the two groups were treated as the OC-Fe measurements. The SOC contents in these fractionations were ultimately determined using an elemental analyzer (Multi EA 4000, Analytik Jena, Germany) after inorganic C was removed with 1 M HCl.

**Soil chemical properties.** The chemical properties of pH, CEC, $Ca^{2+}$, and $Mg^{2+}$ were determined. Soil pH was determined using a pH electrode (Seven Excellence S479-uMix, Mettler-Toledo, Switzerland) in a 1:2.5 soil:water suspension. $Ca^{2+}$ and $Mg^{2+}$ contents were measured by using inductively coupled plasma–optical emission spectrometry[69]. CEC was determined by using a microplate reader (Synergy 2, BioTek, USA) following extraction using $[Co(NH_3)_6]Cl_3$ (ref. [70]).

**Substrate properties.** The substrate quantity included the SOC and SIC contents. The SOC content was analyzed using an elemental analyzer (Multi EA 4000, Analytik Jena, Germany) after inorganic C was removed with 1 M HCl. SIC was determined by a pressure calcimeter method[71]. Briefly, 0.5 g of soil was mixed with 2 mL 6 M HCl and reacted in a closed reaction vessel. Two hours later, the pressure was determined using the pressure transducer and voltage meter, and then the carbonate concentration was calculated using a calibration curve, which was obtained in the same way using known quantities of $CaCO_3$. The SIC content was finally determined by multiplying by a coefficient of 0.12, which is the mass proportion of C in calcium carbonate. The substrate availability was indicated by the C availability index, which was defined as the ratio of the basal respiration to the substrate-induced respiration[24]. A $60\,g\,L^{-1}$ glucose solution was added for the substrate-induced respiration, and deionized water was added in the same manner for the basal respiration rate; respiration rates at 20 °C were estimated for the added glucose and ambient-substrate treatments. The substrate quality was indicated by SOC decomposability ($D_{SOC}$), that is the SOC decomposition rate per unit of SOC content per hour. $D_{SOC}$ was calculated by the ratio of $B$ (the parameter from Eq. (4)) to SOC content.

**Statistical analyses**
A paired-sample $t$-test was applied to examine the differences in $Q_{10\_SOC}$ or $Q_{10\_SIC}$ among different experimental moisture treatments (20%, 40%, and 60% WHC). Correlation analysis was conducted to test the correlations of $Q_{10\_SOC}$ and $Q_{10\_SIC}$ with each variable tested. Additionally, SEM was conducted to partition the direct and indirect effects of climate, physical, chemical, and substrate properties on $Q_{10\_SOC}$ and $Q_{10\_SIC}$. Because the variables within each of these groups were closely correlated, principal component analyses were conducted to create a multivariate functional index prior to SEM analyses[14,72]. The first component was used for the combined group properties in the SEM analysis. The maximum likelihood estimation method was used to fit the data in the SEM analysis. The selection of the final model was based on the $p$-value, $\chi^2$ test, root-mean-squared error of approximation, and goodness-of-fit index[73]. The SEM was conducted using AMOS 21.0 software (Amos Development Corporation, Chicago, IL).

**Reporting summary**
Further information on research design is available in the Nature Portfolio Reporting Summary linked to this article.

## Data availability
Supplementary Information is available online. The $Q_{10}$ value and soil properties data are available at https://doi.org/10.5281/zenodo.10370941.

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

## Acknowledgements

We thank Xinxin Xu for the assistance in soil sampling and laboratory analyses and Hao Liu for the help with the drawing of the distribution map of soil sampling sites. This work was supported by the Science & Technology Fundamental Resources Investigation Program (2023FY100100), the National Natural Science Foundation of China (92251305, 32101377, and 32101336), the Program of Shanghai Academic/Technology Research Leader (21XD1420700), the Science and Technology Plan Project of Shanghai (23DZ1202700), and the Natural Science Foundation of Shanghai (23ZR1404400).

## Author contributions

J.L. and M.N. designed the research. J.L. conducted the overall experiment with the assistance of J.P.; J.L. performed the overall analysis. J.L. wrote the first draft; J.L., M.N., B.L., and C.F. contributed substantially to reviewing and editing the paper.

## Competing interests

The authors declare no competing interests.
