## [Peer Review File · Nature Communications]

REVIEWER COMMENTS

Reviewer #1 (Remarks to the Author):

This research addresses an important question regarding the carbon cycle in drylands and I think that it has been very well carried out. The methods are mostly well explained and the authors have clearly taken a careful and rigorous approach. The findings have important implications which the authors explain well. The interpretation makes sense and is based upon the data presented, however I have got some queries about the detail of the method and how this could impact upon the results.

My main concern with the methods is that soil was sieved and purged with CO₂ free air (line 428). If I understood correctly this was done for both the field based and lab based parts (lines 418, 420). One point of clarity to address is that it was not always clear which of the 2 experiments were being explained, this occurs throughout. I think some work is needed to clearly differentiate these 2 experiments.

Equation (2) shows an equilibrium which will be affected by the amount of CO₂ in the system. In dryland soils the CO₂ concentrations in the pore space are commonly in thousands of ppm but in this experiment the CO₂ has been artificially set at nil. I think that this will cause enhanced SIC-derived CO₂ emissions which could explain one of the main results of the study as being an experimental artefact. A second problem of using sieved soil is that the organisation of the soil which may have built up over years is destroyed and ignored. A 2 week recovery period was mentioned but this is unlikely to be enough for restoring soil properties and natural functioning. Biocrust will have been mixed into the whole soil, dispersing microbes and carbon and eliminating niches in the soil. Fungal hyphae will be broken and no longer able to function properly. There will be therefore microbial death releasing carbon and other resources into the soil. Furthermore mineral exposure will have been altered (e.g. as explained line 222), probably increasing the chance of SIC-derived CO₂ emissions. Thus, the experiment is looking mainly at the disturbance response to sieving which is likely to be dominated by copiotrophs / R strategists which are not representative of natural undisturbed soils.

An in-situ field study could guard against these problems and I initially thought this had been done, but on further reading of this work I concluded that the field study also employed the same approach of sieving - so the results cannot be interpreted as representative of natural soils. This does not totally invalidate the work which I do think has been carried out exceptionally well, but I think that the interpretation needs to be adjusted. The claim on line 74 about using multiple approaches is not completely valid in my view if the same approach to determining the Q₁₀ from sieved soils was used.

In addition, I have a few minor queries and comments:

- L106 and 194 the contribution of SIC-derived emissions is reported to change as a fraction of the total emissions, however I think it would be more clear to say in addition whether the absolute SIC and SOC derived emissions change. The change in proportion could be driven by SOC derived change so it is not clear.

- Sequence data have not been provided for review, nor is an accession number provided. This is essential to provide in my view

- Code availability is kindly offered upon request, I suggest instead to include it as supplementary data instead.

I hope that you found these comments constructive, and that you will be able to improve the already impressive manuscript.

Reviewer #2 (Remarks to the Author):

See attachment

Reviewer #3 (Remarks to the Author):

The paper describes the importance of accounting for the temperature sensitivity of both SOC and SIC in drylands, noting that the temperature sensitivity of SIC may increase under drought. The work is timely, original, significant to the field and the data and methodology are sound. However, the authors have often conflated warming and drying in some of their interpretations, and this led to some confusion throughout the discussion. While warming and drying are often coupled, it is important that the authors define their conclusions by what can be supported by the data. As well, there are some outdated metrics included and the authors might revisit recent literature regarding microbial community composition and function based on life history strategies. There is an impressive amount of data that has gone into this paper, but it has not always been well interpreted. Perhaps the authors might revisit the inclusion of everything that was collected just for the purpose of including it, but instead ask how it addresses hypotheses. In this paper, there is a distinct lack of hypotheses. I do realize that for some scientists, the jury is out on hypothesis-based research, however, considering the number of metrics that were collected, this component of the paper could be far improved. I might suggest there is a paper in the microbial data itself – and this is one piece of the paper where the data was not well interpreted.

I do believe this paper is suitable for the target journal, with significant revisions. I would like to see the authors put some effort into a graphical abstract that demonstrates the directionality of the controls in more detail given the number of metrics they include (Figure 6 is not useful).

In the paper by Lie, Nie et al the authors aims to evaluate the sensitivity of soil organic and inorganic carbon to future warming and changes in moisture in arid soils. The goal of the paper is of major importance as we know very little on the dynamics of SIC, a large carbon reservoir that is often unaccounted for in global carbon cycle models. The authors did an impressive and comprehensive work by studying the effects of aridity on SIC and SOC in top and sub soils that were sampled from 30 sites along an aridity gradient in China and under various moisture conditions in the lab. The scientific methods seem sound and convincing and the results are clearly presented. However, there are several major and minor issues that needs to be addressed before the paper will be ready for publication in Nature Communications. Please see my comments below.

Major comments:

1. The authors found that Q10 SOC increases with increased aridity while Q10 SIC decreased. How can the authors identify if the observed changes in CO₂ d13C is derived from a decline in SOC or a simultaneous increase in SIC emissions? I suspect that the increased SOC accumulation is the major factor behind the increased SIC-CO₂ emissions. Little organic carbon is decomposed, leaving the fraction of mineralized SIC to be higher.

The authors say in the discussion "because high soil moisture usually stimulates SOC decomposition to a much larger extent than SIC dissolution" (L199) – which means that the observed process drives SOC decomposition and hence soil C losses, which, indirectly, increases SIC. There is no direct impact of moisture on SIC??

2. Figure 3- The changes in SOC and SIC derived CO₂ with moisture seems insignificant statistically. Are they?
3. I think a back of the envelope calculation that takes into account the global impact of SIC-CO₂ emissions in a warmer world is required. A simple one.

Specific comments

L32: Specify why low moisture favor SIC accumulation

L33: What is the fraction of SIC in relation to SOC in these ecosystems?

L38-40: The sentence is not clear. 27% of what? Global soil emissions? What is the 70% in high temperature? Please clarify.

L57: Specify how moisture affects Q10SIC

L104: No isotopic fractionation during SOC decomposition that should be accounted for?

L122: How do you know your observations are not the reflection of an increase in SOC increased without any changes to SIC? you are looking at fraction in % not total values. See my major comment below.

L193: Change "technology" to "approach"

L196: I would expect that SIC accumulation will take more time than SOC, am I correct? if so, SIC losses are more dramatic to the soil C reservoirs than SOC? please discuss.

L201: I am a bit confused. Your findings show the opposite - in high aridity SOC will be less susceptible to decomposition and its accumulation in the soil will increase. In this case the role of SIC will be less important.

L207: Citation needed

L210: You mean high temperature-moisture sensitivity, right?

L215-218: how important will it be, in comparison to the net increase in soil C due to the slower SOC decomposition under future aridity?

L251-254: How does soil pH will change with climate change in arid regions? less rain means higher pH as rainfall is acidic? what will be the impact on SIC?

L397-399: The SIC %, pH and temperature changed linearly with the aridity? please provide more details

L405: All the three samples from three locations were grouped together? Clarify

L419: What was measured to test Q10?

L463-468: How SOC was measured? EA?

L492: The microbial measurements were done for all of the samples? incubation and field samples as well?

Response to reviewers' comments on

“Drought exacerbates dryland soil carbon loss from inorganic carbon under warming”

Response to Reviewer #1

Comments to the Author:

[Comment 1] This research addresses an important question regarding the carbon cycle in drylands and I think that it has been very well carried out. The methods are mostly well explained and the authors have clearly taken a careful and rigorous approach. The findings have important implications which the authors explain well. The interpretation makes sense and is based upon the data presented, however I have got some queries about the detail of the method and how this could impact upon the results.

[Response] Thanks for the reviewer's positive assessments and all insightful comments, which have significantly improved the quality of this work. Detailed responses are listed as follows.

[Comment 2] My main concern with the methods is that soil was sieved and purged with CO₂ free air (line 428). If I understood correctly this was done for both the field based and lab based parts (lines 418, 420). One point of clarity to address is that it was not always clear which of the 2 experiments were being explained, this occurs throughout. I think some work is needed to clearly differentiate these 2 experiments.

[Response] Thanks so much for the comment. We have made the two experiments clear throughout the revised manuscript. For example, in the Methods we have added additional descriptions to clearly differentiate these two experiments.

Please see the page 19, lines 315-325: “To test moisture effects on the Q_{10_SOC} and Q_{10_SIC} , we used two experiments: 1) the soils were incubated under field moisture conditions, and the relationship of Q_{10} with the aridity index was tested; and 2) the soils were incubated under some common moisture content conditions of 20%, 40% and 60% WHC, and the difference in Q_{10} among different moisture conditions was tested. For the first experiment, the soils were quickly sieved to 2 mm at 4°C after they were transported to the laboratory to minimize moisture changes and were then incubated to estimate Q_{10} . For the second experiment, soils sieved to 2 mm were adjusted to different moisture conditions of 20%, 40% and 60% WHC by adding deionized water (with pH approximately 7.3) and were then incubated. For soil incubation for both experiments, 50 g of dry-weight fresh soil, with four experimental replicates, was

maintained under field moisture conditions for the first experiment or adjusted to different moisture conditions (20%, 40% and 60% WHC) for the second experiment and incubated in 250 ml jars.”

Moreover, the main results of the opposing moisture effects on Q_{10_SIC} and Q_{10_SOC} in drylands across a wide aridity index and the results of the differential regulating factors were obtained from the first experiment; the second experiment of experimental moisture treatment was conducted to further verify the opposing moisture effects on Q_{10_SIC} and Q_{10_SOC} when only moisture effects. We have made this clear in the revised manuscript.

Please see the pages 4-5, lines 83-88: “In the first experiment, soil moisture regime differences were evaluated by sampling soils from 30 sites across a wide aridity index (ranging from 0.04 to 0.59) along an approximately 4,500 km east–west transect in the drylands of northern China, with large variations in SOC (0.6 to 2.5% in the topsoil (0–10 cm); 0.4 to 1.3% in the subsoil (35–50 cm)) and SIC (0.1 to 0.6% in the topsoil; 0.3 to 1.6% in the subsoil) contents; in this experiment, Q_{10_SOC} and Q_{10_SIC} were determined with field moisture conditions.”

Please also see the page 5, lines 91-96: “In the second experiment, to directly test the effects of only moisture changes on Q_{10_SOC} and Q_{10_SIC} , we conducted a moisture control experiment by incubating soils under different moisture conditions at 20%, 40% and 60% water holding capacity (WHC). To determine the main drivers and their differences associated with variations in Q_{10_SOC} and Q_{10_SIC} that were determined under field moisture conditions along the aridity gradient.”

Finally, following the comments here and below (*[Comment 3]*), we have conducted additional experiments to test the possible effects of CO₂-free air flushing and soil sieving on Q_{10_SOC} and Q_{10_SIC} . Results showed that CO₂-free air flushing and soil sieving did not influence the overall conclusions of this study. Detail responses could be found below (please see the response to *[Comment 3]*).

[Comment 3] Equation (2) shows an equilibrium which will be affected by the amount of CO₂ in the system. In dryland soils the CO₂ concentrations in the pore space are commonly in thousands of ppm but in this experiment the CO₂ has been artificially set at nil. I think that this will cause enhanced SIC-derived CO₂ emissions which could explain one of the main results of the study as being an experimental artefact. A second problem of using sieved soil is that the organisation of the soil which may have built up over years is destroyed and ignored. A 2 week

recovery period was mentioned but this is unlikely to be enough for restoring soil properties and natural functioning. Biocrust will have been mixed into the whole soil, dispersing microbes and carbon and eliminating niches in the soil. Fungal hyphae will be broken and no longer able to function properly. There will be therefore microbial death releasing carbon and other resources into the soil. Furthermore mineral exposure will have been altered (e.g. as explained line 222), probably increasing the chance of SIC-derived CO₂ emissions. Thus, the experiment is looking mainly at the disturbance response to sieving which is likely to be dominated by copiotrophs / R strategists which are not representative of natural undisturbed soils.

[Response] Thanks so much for these comments. Firstly, to directly distinguish the SOC- and SIC-derived CO₂ emissions using the ¹³C natural isotope approach, the soils were flushed with CO₂-free air in the incubation experiments following previous studies (e.g., Schindlbacher et al., 2015; Cardinael et al., 2020). To test the possible effects of flushing with CO₂-free air during soil incubation for Q_{10} estimation, we conducted a supplementary experiment using six representative soils (see Supplementary Text). Results showed that CO₂-free air flushing significantly underestimated SOC- and SIC-derived CO₂ emissions (Supplementary Fig. 6); however, CO₂-free air flushing had no significant effects on Q_{10_SIC} and Q_{10_SOC} (Supplementary Fig. 6). We have added these descriptions and discussions in the revised manuscript.

Please see the page 21, lines 354-363: “Although the soil incubation experiment allowed us to reveal a general pattern and the mechanisms of moisture effects on Q_{10_SIC} and Q_{10_SOC} across large scales, we were aware that there were some possible influences, such as CO₂-free air flushing and soil sieving. To test the possible effects of CO₂-free air flushing on Q_{10_SOC} and Q_{10_SIC} estimation, we conducted a supplementary experiment using six representative soils (see Supplementary Text). The results showed that CO₂-free air flushing significantly underestimated SOC- and SIC-derived CO₂ emissions (Supplementary Fig. 8); this might be because some of the SOC- and SIC-derived CO₂ might go into the pore space of soil, leading to an underestimation of SOC- and SIC-derived CO₂ emissions. However, CO₂-free air flushing had no significant effects on Q_{10_SIC} and Q_{10_SOC} (Supplementary Fig. 8); this might be because Q_{10} is a ratio for the temperature response of CO₂ emissions, resulting in limited effects of CO₂-free air flushing on Q_{10} values.”

Secondly, although soil sieving is usually conducted for incubation experiments to remove roots and gravel, it might cause possible effects. To test the possible effects of soil sieving on SOC- and SIC-derived CO₂ emissions and their temperature sensitivities, we conducted another

supplementary experiment using non-sieving and sieving soils. Results showed that sieving did not exert significant effects on SOC- and SIC-derived CO₂ emissions and their Q_{10} values (Supplementary Fig. 9). This might be because the soils in drylands are usually sandy, resulting in limited effects from sieving. We have added these descriptions and discussions in the revised manuscript.

Please see the page 21, lines 363-368: “Moreover, we conducted another supplementary experiment using intact and sieved soils to test the effects of soil sieving on Q_{10_SOC} and Q_{10_SIC} (see Supplementary Text). The results showed that sieving did not exert significant effects on SOC- and SIC-derived CO₂ emissions and their Q_{10} values (Supplementary Fig. 9). This might be because the soils in drylands are usually sandy, resulting in limited effects from sieving. A recent study has also shown that soil sieving had no significant effects on the Q_{10} of total CO₂ emissions⁶⁵.”

Finally, we based on the comments here and that from Reviewer #3 (*[Comment 19]* and *[Comment 24]*), we have removed the use of metric oligotroph:copiotroph ratio in the revised manuscript.

[Comment 4] An in-situ field study could guard against these problems and I initially thought this had been done, but on further reading of this work I concluded that the field study also employed the same approach of sieving - so the results cannot be interpreted as representative of natural soils. This does not totally invalidate the work which I do think has been carried out exceptionally well, but I think that the interpretation needs to be adjusted. The claim on line 74 about using multiple approaches is not completely valid in my view if the same approach to determining the Q_{10} from sieved soils was used.

[Response] Thanks so much the comment. Following the comment, we have weakened the representative of natural soils especially in the Results section, while we have inferred in some places to highlight the potential implications in the Discussion section in the revised manuscript. For example, please see the pages 13-14, lines 205-208: “Our results thus indicate that the dissolution of SIC is more important than previously thought in regulating atmospheric CO₂ concentrations⁸, and if future climate change accelerates aridity in drylands²⁰, the contribution of SIC-derived CO₂ to total CO₂ emissions may become even more significant.”

Please also see the page 16, lines 264-266: “In conclusion, our findings provide evidence for opposing moisture effects on the temperature response of SOC and SIC in drylands, suggesting

that drying will further enhance the temperature response of SIC but weaken that of SOC (Fig. 6).”

Secondly, we have removed the statement of using multiple approaches. Instead, we have stated conducting two experiments in the revised manuscript (for example, please see the page 4, lines 82-83).

Finally, following the comments here and above (*Comment 3*), we have conducted an additional experiment to reveal the potential effects of soil sieving on SOC- and SIC-derived CO₂ emissions and their Q_{10} values and have added some discussions regarding this in the revised manuscript.

Please also see the page 21, lines 363-368: “Moreover, we conducted another supplementary experiment using intact and sieved soils to test the effects of soil sieving on Q_{10_SOC} and Q_{10_SIC} (see Supplementary Text). The results showed that sieving did not exert significant effects on SOC- and SIC-derived CO₂ emissions and their Q_{10} values (Supplementary Fig. 9). This might be because the soils in drylands are usually sandy, resulting in limited effects from sieving. A recent study has also shown that soil sieving had no significant effects on the Q_{10} of total CO₂ emissions⁶⁵.”

Specific comments:

[Comment 5] L106 and 194 the contribution of SIC-derived emissions is reported to change as a fraction of the total emissions, however I think it would be more clear to say in addition whether the absolute SIC and SOC derived emissions change. The change in proportion could be driven by SOC derived change so it is not clear.

[Response] Thanks so much for the comment. Following the comment here and that from Reviewer #2 (*Comment 2*) and (*Comment 10*), we have conducted additional analysis to reveal the changes of absolute SOC- and SIC-derived CO₂ emissions with soil moisture changes along the aridity gradient, showing that SOC- and SIC-derived CO₂ both decreased with increasing aridity (Supplementary Fig. 2). Similar results were observed from the moisture control experiment, showing that SOC- and SIC-derived CO₂ rates were lower under lower experimental moisture conditions (Supplementary Fig. 3). We have added these analyses in the revised manuscript (please see the page 7, lines 118-121).

Supplementary Fig. 2 | Changes in SOC-derived and SIC-derived CO₂ emissions in the topsoil along an aridity gradient. a, Linear relationships of SOC-derived CO₂ with the aridity index. **b**, Linear relationships of SIC-derived CO₂ with the aridity index. The dashed lines correspond to the 95% confidence interval of the linear relationship. SOC-derived and SIC-derived CO₂ emissions were estimated under field moisture conditions at 20°C.

Supplementary Fig. 3 | Differences in SOC- and SIC-derived CO₂ emissions in the topsoil among different experimental moisture treatments. a, Differences in SOC-derived CO₂ among different moisture conditions. **c**, Differences in SIC-derived CO₂ among different moisture conditions. The whiskers illustrate standard deviations ($n = 30$), the ends of the boxes indicate the 25th and 75th percentiles, and the horizontal lines inside each box represent the

median. The gray dots indicate values for each of the 30 sites. SOC- and SIC-derived CO₂ emissions were estimated at 20°C. *, $P < 0.05$; ***, $P < 0.001$; WHC, water holding capacity.

[Comment 6] Sequence data have not been provided for review, nor is an accession number provided. This is essential to provide in my view.

[Response] Thanks so much for the comments. Based on the comments (*[Comment 1]*, *[Comment 19]* and *[Comment 24]*) from Reviewer #3 and revisiting the recent study (Stone *et al.*, 2023, *ISME*), we have realized that soil microbial life history (the copiotroph-oligotroph framework) classification based on the phylum level like our study would be problematic. We thus have removed the use of metric oligotroph:copiotroph ratio representing microbial life history in the revised manuscript. We have still included bacterial diversity and fungal diversity data to represent microbial properties. We have uploaded the sequence data into NCBI (<https://www.ncbi.nlm.nih.gov/bioproject/PRJNA990330/>), and we have noted this in the revised manuscript (please see the page 24, lines 438-440).

[Comment 7] Code availability is kindly offered upon request, I suggest instead to include it as supplementary data instead.

[Response] Thanks so much for the comments. We have made the data of soil physical properties, chemical properties, substrate and microbial properties public available (<http://doi.org/10.5281/zenodo.8109607>). We made the statement “*Code availability is kindly offered upon request*”, which is often written by the journal Nature Communications, in case that readers want some special data that we were negligent. We have made this statement with the purpose to better communicate with readers. Thanks for your understanding.

[Comment 8] I hope that you found these comments constructive, and that you will be able to improve the already impressive manuscript.

[Response] Thanks so much for these insightful comments, which have significantly improved the quality of this work.

Response to Reviewer 2

Comments to the Author:

[Comment 1] In the paper by Lie, Nie et al the authors aims to evaluate the sensitivity of soil organic and inorganic carbon to future warming and changes in moisture in arid soils. The goal of the paper is of major importance as we know very little on the dynamics of SIC, a large carbon reservoir that is often unaccounted for in global carbon cycle models. The authors did an impressive and comprehensive work by studying the effects of aridity on SIC and SOC in top and sub soils that were sampled from 30 sites along an aridity gradient in China and under various moisture conditions in the lab. The scientific methods seem sound and convincing and the results are clearly presented. However, there are several major and minor issues that needs to be addressed before the paper will be ready for publication in Nature Communications. Please see my comments below.

[Response] Thanks for the reviewer's positive assessments and all insightful comments, which have significantly improved the quality of this work. Detailed responses are listed as follows.

[Comment 2] The authors found that Q10 SOC increases with increased aridity while Q10 SIC decreased. How can the authors identify if the observed changes in CO₂ d13C is derived from a decline in SOC or a simultaneous increase in SIC emissions? I suspect that the increased SOC accumulation is the major factor behind the increased SIC-CO₂ emissions. Little organic carbon is decomposed, leaving the fraction of mineralized SIC to be higher.

The authors say in the discussion "because high soil moisture usually stimulates SOC decomposition to a much larger extent than SIC dissolution" (L199) – which means that the observed process drives SOC decomposition and hence soil C losses, which, indirectly, increases SIC. There is no direct impact of moisture on SIC??

[Response] Thanks so much for the comment. Based on the comment here and below (JComment 10J) and that from Reviewer #1 ((JComment 5J), we have conducted additional analyses regarding the changes in absolute SOC- and SIC-derived CO₂ emissions. Results showed that SOC- and SIC-derived CO₂ both decreased with increasing aridity (Supplementary Fig. 2). Similar results were observed from the moisture control experiment, showing that SOC- and SIC-derived CO₂ rates were lower under lower experimental moisture conditions (Supplementary Fig. 3). We have added these analyses in the revised manuscript (please see the page 7, lines 118-121).

Therefore, we can infer that soil moisture has direct effects on both SOC and SIC processes. We have also revised previous statement of "Moreover, we showed that the contribution of SIC to CO₂ emissions increased with decreasing moisture along the aridity gradient and the experimental moisture treatments because high soil moisture usually stimulates SOC decomposition to a much larger extent than SIC dissolution." in the revised manuscript.

Please see the page 13, lines 201-205: "Moreover, we observed that absolute SOC- and SIC-derived CO₂ emissions both decreased, but the contribution of SIC to total CO₂ emissions increased with decreasing moisture along the aridity gradient and the experimental moisture treatments; this might be because the microbial process of SOC decomposition is more sensitive to moisture changes than the chemical process of SIC dissolution^{17,41}."

As absolute SOC- and SIC-derived CO₂ emissions both changes with moisture changes, we thus can speculate that both Q_{10_SOC} and Q_{10_SIC} change in response to moisture changes along the aridity gradient.

[Comment 3] Figure 3- The changes in SOC and SIC derived CO₂ with moisture seems insignificant statistically. Are they?

[Response] Thanks for the comment. We have conducted additional analyses to test the differences of SOC- and SIC-derived CO₂ among different moisture conditions. Results showed that SOC- and SIC-derived CO₂ rates were lower under lower experimental moisture conditions (Supplementary Fig. 3). We have added these results in the revised manuscript (please see the page 7, lines 118-121).

Supplementary Fig. 3 | Differences in SOC- and SIC-derived CO₂ emissions in the topsoil among different experimental moisture treatments. **a**, Differences in SOC-derived CO₂ among different moisture conditions. **c**, Differences in SIC-derived CO₂ among different moisture conditions. The whiskers illustrate standard deviations ($n = 30$), the ends of the boxes indicate the 25th and 75th percentiles, and the horizontal lines inside each box represent the median. The gray dots indicate values for each of the 30 sites. SOC- and SIC-derived CO₂ emissions were estimated at 20°C. *, $P < 0.05$; ***, $P < 0.001$; WHC, water holding capacity.

[Comment 4] I think a back of the envelope calculation that takes into account the global impact of SIC-CO₂ emissions in a warmer world is required. A simple one.

[Response] Thanks so much for the comment. We have added an envelope calculation on the impact of warming on SIC-derived CO₂ emissions in the revised manuscript.

Please see the page 14, lines 223-228: “Total SOC- and SIC-derived CO₂ emissions are roughly estimated at 20.4 Pg C year⁻¹ in drylands (assuming that soil respiration in drylands accounts for 38.6% of global soil respiration⁴⁵, 60% of which is from the heterotrophic component⁴⁶), and SIC-derived CO₂ contributes to approximately 27.0% of total CO₂ emissions¹⁶. Using Q_{10_SOC} to represent Q_{10_SIC} would underestimate warming-induced SIC-derived CO₂ emissions (3.2 Pg C year⁻¹) by approximately 25.6% compared to that estimated (4.3 Pg C year⁻¹) using the higher Q_{10_SIC} under 4°C of warming.”

Specific comments:

[Comment 5] L32: Specify why low moisture favor SIC accumulation.

[Response] Thanks for the comment. Based on the comment here and that from Reviewer #3 (*[Comment 3]*), we have revised this sentence in the revised manuscript.

Please see the page 2, lines 30-33: “The presence of soil inorganic carbon (SIC) is primarily regulated by parent material⁸, and the low moisture conditions in drylands favor a higher ratio of SIC to soil organic carbon (SOC) given that plant inputs are low, with approximately 2–10 times more SIC storage than SOC in these ecosystems⁹.”

[Comment 6] L33: What is the fraction of SIC in relation to SOC in these ecosystems?

[Response] SIC content is about 2–10 times more SIC storage than soil organic carbon (SOC) in dryland ecosystems. We have added this in the revised manuscript (**please see the page 2, line 33**).

[Comment 7] L38-40: The sentence is not clear. 27% of what? Global soil emissions? What is the 70% in high temperature? Please clarify.

[Response] Thanks so much for the comment. We have made these clear in the revised manuscript.

Please see the page 3, lines 38-40: “A recent synthesis of 28 studies has shown that SIC-derived CO₂ contributed 27% of total CO₂ emissions from calcareous soils¹⁶, and the contribution can reach 70% at high temperatures in some specific cases¹⁸.”

[Comment 8] L57: Specify how moisture affects Q10SIC.

[Response] Thanks for the comment. We have added some more details on the potential effects of moisture on Q_{10_SIC} here in the revised manuscript.

Please see the pages 3-4, lines 58-66: “For example, a decrease in soil moisture would inhibit CaCO₃ dissolution¹⁶, and the observed net SIC-derived CO₂ emissions can be especially low under low moisture conditions, as CO₂ is also consumed during carbonate dissolution³¹. Soil moisture can also indirectly affect SIC dissolution by mediating soil pH and/or base cations (e.g., Ca²⁺ and Mg²⁺)¹⁶ that can shift the reactions represented in Equations (1) and (2). Accordingly, a drop in soil H⁺ owing to the enhancement of soil pH resulting from soil moisture decrease³² will lead the reactions represented in Equations (1) and (2) to proceed to the left, leading to low SIC-derived CO₂ emission rates. Given that a low CO₂ emission rate is more sensitive to environmental changes (e.g., temperature), moisture decrease and/or pH or base cation increase may enhance the temperature response of SIC dissolution.”

[Comment 9] L104: No isotopic fractionation during SOC decomposition that should be accounted for?

[Response] Yes, there is isotopic fractionation during SOC decomposition, and we have accounted for as described in the Methods (please see the page 20, lines 344-345). Following the comment, to be more accurate statement, we have also added descriptions here (please see the page 7, lines 112-114).

[Comment 10] L122: How do you know your observations are not the reflection of an increase in SOC increased without any changes to SIC? you are looking at fraction in % not total values. See my major comment above.

[Response] Thanks so much for the comments. Following the comments here and above (*JComment 2J*) and that from Reviewer #1 (*JComment 5J*), we have conducted additional analyses regarding the changes in absolute SOC- and SIC-derived CO₂ emissions. Results showed that SOC- and SIC-derived CO₂ both decreased with increasing aridity (Supplementary Fig. 2). We have added these analyses in the revised manuscript (please see the page 7, lines 118-121). Detailed responses could also be found in the response to above comment (*JComment 2J*).

[Comment 11] L193: Change "technology" to "approach".

[Response] Done as suggested (please see the page 13, line 197).

[Comment 12] L196: I would expect that SIC accumulation will take more time than SOC, am I correct? if so, SIC losses are more dramatic to the soil C reservoirs than SOC? please discuss.

[Response] Yes, we agree that SIC accumulation takes significantly more time than SOC. We have added some discussions on this in the revised manuscript.

Please see the page 13, lines 199-201: “Given that SIC accumulation usually takes significantly more time than SOC⁴⁰, SIC losses are thus more impactful for the soil C reservoirs than that of SOC in these water-limited ecosystems.”

[Comment 13] L201: I am a bit confused. Your findings show the opposite - in high aridity SOC will be less susceptible to decomposition and its accumulation in the soil will increase. In this case the role of SIC will be less important.

[Response] Sorry for the confusion. We have revised this sentence as “if future climate change accelerates aridity in drylands²¹, the contribution of SIC-derived CO₂ to total CO₂ emissions will become even more significant” in the revised manuscript (please see the pages 13-14, lines 206-208). Because our results showed that the contribution of SIC to CO₂ emissions increased with decreasing moisture along the aridity gradient and the experimental moisture treatments.

[Comment 14] L207: Citation needed.

[Response] Done as suggested (please see the page 14, line 213).

[Comment 15] L210: You mean high temperature-moisture sensitivity, right?

[Response] Yes, we have revised it as “the high temperature-moisture sensitivity of SIC” in the revised manuscript (please see the page 14, line 216).

[Comment 16] L215-218: how important will it be, in comparison to the net increase in soil C due to the slower SOC decomposition under future aridity?

[Response] Thanks so much for the comment. Based on the comment here and that from above (*Comment 4*), we have added envelope calculations on the impact of warming on SIC-derived CO₂ emissions and its implication considering to the slower SOC decomposition under future aridity in the revised manuscript.

Please see the pages 14-15, lines 223-234: “Total SOC- and SIC-derived CO₂ emissions are roughly estimated at 20.4 Pg C year⁻¹ in drylands (assuming that soil respiration in drylands accounts for 38.6% of global soil respiration⁴⁵, 60% of which is from the heterotrophic component⁴⁶), and SIC-derived CO₂ contributes to approximately 27.0% of total CO₂ emissions¹⁶. Using Q_{10_SOC} to represent Q_{10_SIC} would underestimate warming-induced SIC-derived CO₂ emissions (3.2 Pg C year⁻¹) by approximately 25.6% compared to that estimated (4.3 Pg C year⁻¹) using the higher Q_{10_SIC} under 4°C of warming. Moreover, considering the different responses of Q_{10} to moisture changes (Q_{10_SOC} decreases by 0.47 and Q_{10_SIC} increases by 0.39 per 0.1 decrease in the aridity index; **Fig. 2**), the net increase in soil C due to the lower Q_{10_SOC} (1.5 Pg C year⁻¹) would be offset by approximately 26.7% due to the higher Q_{10_SIC} (0.4 Pg C year⁻¹) under 0.1 decreases in the aridity index. Consequently, although these are rough estimates, they highlight the importance of separately representing Q_{10_SOC} and Q_{10_SIC} and their different responses to moisture changes to improve projections of climate–C cycle feedbacks in drylands.”

[Comment 17] L251-254: How does soil pH will change with climate change in arid regions? less rain means higher pH as rainfall is acidic? what will be the impact on SIC?

[Response] Thanks so much for the comment. Drying is generally predicted to increase soil pH, which will further enhance the temperature response of SIC. We have added these discussions in the revised manuscript (please see the page 16, lines 259-260).

[Comment 18] L397-399: The SIC %, pH and temperature changed linearly with the aridity? Please provide more details.

[Response] Thanks so much for the comment. We have added additional analyses regarding the relationships of SIC, pH and temperature with aridity. Results showed that pH and SIC content were significant correlated with aridity (Supplementary Fig. 7), while no significant

relationship between mean annual temperature and aridity were observed along the aridity gradient (Supplementary Fig. 6). We have incorporated these results in the revised manuscript. Please see the page 18, lines 294-299: “The MAT and MAP ranged from -1.2 to 10.0°C and from 46 to 486 mm, respectively, resulting in a wide range of aridity, with aridity indices (the ratio of precipitation to potential evapotranspiration) ranging from 0.04 to 0.59 at these sites; no significant relationship between MAT and AI was observed along the aridity gradient (Supplementary Fig. 6). Soil properties for these sites were also greatly different; for example, soil pH and SIC content were significantly and negatively correlated with AI (Supplementary Fig. 7).”

Supplementary Fig. 6 | Relationships of MAT with the aridity index along an aridity gradient in drylands. MAT, mean annual temperature.

Supplementary Fig. 7 | Changes in soil pH and SIC content along an aridity gradient in drylands. a–b, Relationships of soil pH with aridity index in the topsoil (a) and subsoil (b). c–d, Relationships of SIC content with aridity index in the topsoil (c) and subsoil (d).

[Comment 19] L405: All the three samples from three locations were grouped together? Clarify.

[Response] Yes, soils from the three random locations were gently mixed to produce a homogeneous composite sample for each depth at each site. We have added this information in the revised manuscript (please see the page 18, lines 305-307).

[Comment 20] L419: What was measured to test Q10?

[Response] Thanks for the comment. For the first experiment, aridity index was obtained to test its relationship with Q_{10} ; for the second experiment, different moisture content conditions

was obtained and the difference of Q_{10} among different moisture conditions was tested. We have added this information in the revised manuscript.

Please see the page 19, lines 315-318: “To test moisture effects on the Q_{10_SOC} and Q_{10_SIC} , we used two experiments: 1) the soils were incubated under field moisture conditions, and the relationship of Q_{10} with the aridity index was tested; and 2) the soils were incubated under some common moisture content conditions of 20%, 40% and 60% WHC, and the difference in Q_{10} among different moisture conditions was tested.”

[Comment 21] L463-468: How SOC was measured? EA?

[Response] SOC content was determined using an elemental analyzer (Multi EA 4000, Analytik Jena, Germany) after inorganic C was removed with 1 M HCl. We have noted these descriptions in the revised manuscript (please see the page 22, lines 386-388).

[Comment 22] L492: The microbial measurements were done for all of the samples? incubation and field samples as well?

[Response] The microbial measurements were done for the 60 soil samples (30 sites \times 2 soil depths) collected across the natural aridity gradient. Because the mechanism determination was conducted for the the variations in the Q_{10} of the SOC- and SIC-derived CO₂ emissions along the aridity gradient. We have noted this in the revised manuscript (please see the page 5, lines 94-100; please also see the pages 21-22, lines 374-377).

Response to Reviewer 3

Comments to the Author:

[Comment 1] The paper describes the importance of accounting for the temperature sensitivity of both SOC and SIC in drylands, noting that the temperature sensitivity of SIC may increase under drought. The work is timely, original, significant to the field and the data and methodology are sound. However, the authors have often conflated warming and drying in some of their interpretations, and this led to some confusion throughout the discussion. While warming and drying are often coupled, it is important that the authors define their conclusions by what can be supported by the data. As well, there are some outdated metrics included and the authors might revisit recent literature regarding microbial community composition and function based on life history strategies. There is an impressive amount of data that has gone into this paper, but it has not always been well interpreted. Perhaps the authors might revisit the inclusion of everything that was collected just for the purpose of including it, but instead ask how it addresses hypotheses. In this paper, there is a distinct lack of hypotheses. I do realize that for some scientists, the jury is out on hypothesis-based research, however, considering the number of metrics that were collected, this component of the paper could be far improved. I might suggest there is a paper in the microbial data itself – and this is one piece of the paper where the data was not well interpreted.

[Response] Thanks for the reviewer's positive assessments and all insightful comments, which have significantly improved the quality of this work.

Firstly, we have revised the Discussion section trying to distinguish the discussions/conclusions derived from warming or derived from drying in the revised manuscript (please see the revised the Discussion section).

Secondly, we have removed the use of some outdated metrics in the revised manuscript, including the substrate quality of humification index and the ratio of recalcitrant to labile SOC and microbial life history strategy indicated by the ratio of oligotroph to copiotroph and the ratio of functional genes involving recalcitrant to labile carbon decomposition.

Thirdly, we have added measurements and/or calculations of new metrics of substrate, including carbon availability index (please see the page 23, lines 403-407) and SOC

decomposability (please see the page 23, lines 407-409) that could directly represent the biologically relevant substrate status in the revised manuscript. We have also added measurements of enzyme activities (please see the pages 23-24, lines 417-422) that could directly represent microbial properties in the revised manuscript.

Please see the page 23, lines 403-409: “The substrate availability was indicated by the carbon availability index (CAI), which was defined as the ratio of the basal respiration rate to the substrate-induced respiration ratio²⁶. A 60 g L⁻¹ glucose solution was added for the substrate-induced respiration, and deionized water was added in the same manner for the basal respiration rate; respiration rates at 20°C were estimated for the added-glucose and ambient-substrate treatments. The substrate quality was indicated by SOC decomposability (D_{SOC}), that is, the SOC decomposition rate per unit SOC content per hour. D_{SOC} was calculated by the ratio of B (the parameter from Equation (5)) to SOC content.”

Please also see the pages 23-24, lines 417-422: “Moreover, we also estimated the potential activities of four hydrolytic extracellular enzymes involved in soil C (β -D-cellubiosidase, β -glucosidase, α -glucosidase, β -xylosidase) cycling. The activities of the hydrolytic enzymes were determined following a high-throughput fluorometric measurement protocol⁷³. The total hydrolase activity was summed as the activities of the four hydrolases. Oxidase was measured by spectrophotometry with l-3,4-dihydroxy-phenylalanine as a substrate⁷³.”

Finally, we have revisited the inclusion of potential factors regulating Q_{10} and have added hypotheses in the revised manuscript to help readers better follow this study.

Please see the page 4, lines 80-82: “Here, we hypothesized that (i) Q_{10_SOC} decreased but Q_{10_SIC} increased with decreasing moisture content, and (ii) Q_{10_SOC} was mainly regulated by physicochemical protection, while Q_{10_SIC} was primarily regulated by chemical properties (e.g., pH and CEC).”

Other detailed responses are listed as follows.

[Comment 2] I do believe this paper is suitable for the target journal, with significant revisions. I would like to see the authors put some effort into a graphical abstract that demonstrates the directionality of the controls in more detail given the number of metrics they include (Figure 6 is not useful).

[Response] Thanks so much for the comment. Based on the comments here and below (*[Comment 28]*), we have largely improved the graphical abstract (Figure 6) to demonstrate the

directionality of the controls in more detail in the revised manuscript (please see the new Figure 6).

Fig. 6 | Conceptual diagram showing the differential responses and controls of the temperature sensitivity (Q_{10}) of SOC and SIC to aridity changes in dryland ecosystems. SOC temperature sensitivity (Q_{10_SOC}) decreases with increasing aridity; this is mainly attributed to the increases in mineral protection and/or decreases in substrate availability. However, SIC temperature sensitivity (Q_{10_SIC}) increases with increasing aridity, which is mainly attributed to the increases in soil pH and/or base cations. Notably, this conceptual diagram mainly focused on the most important factors derived from the structural equation modeling analysis.

Specific comments:

[Comment 3] L32: The presence of SIC is primarily regulated by parent material, SIC accumulation can be exacerbated by increased moisture and inputs of exogenic bicarbonates - It is perhaps better to frame these statements differently. I.e. "the low moisture conditions in drylands favor a higher ratio of SIC to SOC given that plant inputs of DOC are low".

[Response] Thanks for the comment. Based on the comment here and that from Reviewer #2 (*Comment 5*), we have revised this sentence in the revised manuscript.

Please see the page 2, lines 30-33: “The presence of soil inorganic carbon (SIC) is primarily regulated by parent material⁸, and the low moisture conditions in drylands favor a higher ratio of SIC to soil organic carbon (SOC) given that plant inputs are low, with approximately 2–10 times more SIC storage than SOC in these ecosystems⁹.”

[Comment 4] L42: Please include some review in this section about how some dryland regions are becoming increasingly prone to flooding, and how this may influence both SIC accumulation and carbonate dissolution and reprecipitation. This would much better encompass the predicted influences of climate change on drylands with regards to SOC-SIC.

[Response] Thanks so much for the comment. We have added some discussions regarding flooding and its potential effects on SOC and SIC in dryland regions in the revised manuscript.

Please see the page 16, lines 260-263: “ Although an overall increase in aridity is predicted in drylands in a warmer world²⁰, some dryland regions are becoming increasingly prone to flooding⁵², leading to increases in pedogenic carbonate accumulation⁵³; however, flooding may also result in losses of both SOC and SIC through soil erosion in these areas⁵⁴.”

[Comment 5] L57: Expand on these processes - SIC dissolution is not linear nor stagnant and reprecipitation processes need some paying attention to.

[Response] Done as suggested.

Please see the pages 3-4, lines 56-66: “Soil moisture changes can directly drive the $\text{CaCO}_3\text{-CO}_2\text{-HCO}_3^-$ equilibrium equations to promote or inhibit CaCO_3 dissolution^{17,30}, indicating that moisture effects on SIC dissolution are not linear or stagnant and that reprecipitation processes may also occur. For example, a decrease in soil moisture would inhibit CaCO_3 dissolution¹⁶, and the observed net SIC-derived CO_2 emissions can be especially low under low moisture conditions, as CO_2 is also consumed during carbonate dissolution³¹. Soil moisture can also indirectly affect SIC dissolution by mediating soil pH and/or base cations (e.g., Ca^{2+} and Mg^{2+})¹⁶ that can shift the reactions represented in Equations (1) and (2). Accordingly, a drop in soil H^+ owing to the enhancement of soil pH resulting from soil moisture decrease³² will lead the reactions represented in Equations (1) and (2) to proceed to the left, leading to low SIC-derived CO_2 emission rates. Given that a low CO_2 emission rate is more sensitive to environmental changes (e.g., temperature), moisture decrease and/or pH or base cation increase may enhance the temperature response of SIC dissolution.”

[Comment 6] L59-60: Consider re-writing this sentence which is confusing because of the multiple uses of 'low' and 'lower' consider low and reduced.

[Response] Thanks so much for the comment. Based on the comments here and above (*Comment 6*), we have largely revised these sentences (please see the pages 3-4, lines 56-66).

[Comment 7] L62: Expand.

[Response] Done as suggested.

Please see the page 4, lines 62-66: “Accordingly, a drop in soil H^+ owing to the enhancement of soil pH resulting from soil moisture decrease³² will lead the reactions represented in Equations (1) and (2) to proceed to the left, leading to low SIC-derived CO_2 emission rates. Given that a low CO_2 emission rate is more sensitive to environmental changes (e.g., temperature), moisture decrease and/or pH or base cation increase may enhance the temperature response of SIC dissolution.”

[Comment 8] L64: If there is no reference for this, please expand describing the processes of the SOC-SIC interaction. Indeed, in calcareous soils, SIC dissolution and reprecipitation is mediated largely by the partial pressure of CO_2 in pore spaces.

[Response] Thanks for the comment. We have added some descriptions on this in the revised manuscript.

Please see the page 4, lines 69-72: “An increase in SOC decomposition will lead to more active CO_2 sources in soil for HCO_3^- production, which will restrict $CaCO_3$ dissolution. Consequently, soil physicochemical protection and microbial communities can regulate SIC processes by mediating SOC decomposition.”

[Comment 9] L68-69: It is my understanding that even this study is constrained to China?

[Response] Thanks for the comment. We have revised it as “across a large moisture gradient” in the revised manuscript (please see the page 4, line 76). Moreover, we have noted that our study area is located in China by stating “along an approximately 4,500 km east–west transect in the drylands of northern China” in the revised manuscript (please see the page 5, lines 85-86).

[Comment 10] L74: To my mind this is not a hypothesis based study, but almost a survey approach. If the authors want to make this the former then the hypothesis-driven nature of the

study needs more attention. Using a large suite of metrics to answer one question suggests there should be more than one directional hypothesis.

[Response] Thanks so much for the comment. Based on the comments here and previous (*Comment 1*), we have added hypotheses in the revised manuscript to help readers better follow this study.

Please see the page 4, lines 80-83: “Here, we hypothesized that (i) Q_{10_SOC} decreased but Q_{10_SIC} increased with decreasing moisture content, and (ii) Q_{10_SOC} was mainly regulated by physicochemical protection, while Q_{10_SIC} was primarily regulated by chemical properties (e.g., pH and CEC). To test these hypotheses, we conducted two experiments (**Fig. 1**): a natural aridity gradient and a moisture control treatment.”

[Comment 11] L77: Please explain the range of SOC and SIC contents throughout the study areas.

[Response] Done as suggested.

Please see the pages 4-5, lines 83-87: “In the first experiment, soil moisture regime differences were evaluated by sampling soils from 30 sites across a wide aridity index (ranging from 0.04 to 0.59) along an approximately 4,500 km east–west transect in the drylands of northern China, with large variations in SOC (0.6 to 2.5% in the topsoil (0–10 cm); 0.4 to 1.3% in the subsoil (35–50 cm)) and SIC (0.1 to 0.6% in the topsoil; 0.3 to 1.6% in the subsoil) contents.”

[Comment 12] L86: An extremely important mechanism that has not been properly introduced until now. Needs attention in the introduction.

[Response] Thanks for the comment. We have added some introduction in the revised manuscript.

Please see the page 4, lines 70-73: “Consequently, soil physicochemical protection and microbial communities can regulate SIC processes by mediating SOC decomposition. Although mineral protection of Ca bridges and/or Fe oxides has been shown to largely inhibit SOC decomposition and its temperature sensitivity^{14,34}, its effects on Q_{10_SIC} remain unknown.”

[Comment 13] L96: New sentence.

[Response] Done as suggested (please see the page 6, line 106).

[Comment 14] L99: In the conceptual diagram the authors somewhat hypothesize a directionality difference. So far this study is lacking hypothetical directionality. It is not novel

to just say 'these things will be different' especially considering they acknowledge the different physicochemical drivers throughout the introduction.

[Response] Thanks so much for the comment. Following the comments here and previous (*Comment 1*) and (*Comment 10*), we have added hypotheses in the revised manuscript.

Please see the page 4, lines 80-83: “Here, we hypothesized that (i) Q_{10_SOC} decreased but Q_{10_SIC} increased with decreasing moisture content, and (ii) Q_{10_SOC} was mainly regulated by physicochemical protection, while Q_{10_SIC} was primarily regulated by chemical properties (e.g., pH and CEC). To test these hypotheses, we conducted two experiments (**Fig. 1**): a natural aridity gradient and a moisture control treatment.”

[Comment 15] L121: Remove “increasing aridity index”.

[Response] Done as suggested.

[Comment 16] L122: Just state the ecological importance the metric is clear from the figure.

[Response] Thanks for the comment. We have changed the statement to highlight the ecological importance of drying in the revised manuscript.

Please see the page 8, lines 131-133: “Opposing patterns of Q_{10_SOC} and Q_{10_SIC} were observed in response to aridity changes, showing that in both soil layers, Q_{10_SOC} decreased significantly with drying (that is, decreasing aridity index), whereas Q_{10_SIC} increased significantly with drying ($P < 0.05$, **Fig. 2**).”

[Comment 17] L124: Given that the authors are concerned with climate change impacts, and state overall that aridity will increase, state the importance of having chosen to test these levels.

[Response] Thanks for the comment. We have stated the importance of having chosen these moisture levels in the revised manuscript.

Please see the page 8, lines 133-135: “This was further verified by the moisture control experiment; given the predicted overall aridity increase in drylands in a warmer world, soils were incubated under different moisture contents of 20%, 40% and 60% WHC.”

[Comment 18] L131: Given there was no interactive effects (moisture:depth) the authors might consider faceting these graphs differently to make this clearer.

[Response] Thanks so much for the comment. Following the comment, we have re-drawn the Figure 3 in the revised manuscript (**please see the new Figure 3**).

Fig. 3 | Differences in the temperature sensitivity (Q_{10}) of SOC- and SIC-derived CO_2 emissions among different experimental moisture treatments. a–b, Differences in the Q_{10} of SOC-derived CO_2 in the topsoil (a) and subsoil (b) among different moisture conditions. c–d, Differences in the Q_{10} of SIC-derived CO_2 in the topsoil (c) and subsoil (d) among different moisture conditions. The horizontal lines inside the box represent the median, and the ends of the boxes represent the 25th and 75th percentiles. The gray dots indicate values for each of the 30 sites. **, $P < 0.01$; *, $P < 0.001$; WHC, water holding capacity.**

[Comment 19] L143: The authors may want to revisit their interpretation of these ratios given recent experimental evidence suggesting that many of these ratios are arbitrary at best, and not well-defined <https://www.nature.com/articles/s41396-022-01354-0>.

[Response] Thanks so much for the comment. Based on the comment and revisiting the recent study (Stone *et al.*, 2023, *ISME*), we have realized that soil microbial life history (the

copiotroph-oligotroph framework) classification based on the phylum level like our study would be problematic. Therefore, we have removed the use of metric oligotroph:copiotroph ratio in the revised manuscript.

[Comment 20] L147: Remove “However”.

[Response] Done as suggested.

[Comment 21] Figure 5: Insignificant substrate effects should be expanded upon in discussion. As suggested, the authors may want to reconsider the use of the 'humification index', or at least frame the non-significance in the context of methodology.

[Response] Thanks so much for the comment. Based on the comments here, above (*[Comment 1]*) and below (*[Comment 22]* and *[Comment 32]*), we have removed the metric of humification index in the revised manuscript. We have added measurements and/or calculations of new metrics of substrate, including carbon availability index (please see the page 23, lines 403-407) and SOC decomposability (please see the page 23, lines 407-409) that could directly represent the biologically relevant substrate status in the revised manuscript. The new results showed that substrate had significant direct effects on Q_{10_SOC} (please see **Fig. 5a**). We have added some discussions on this in the revised manuscript.

Please see the page 15, lines 245-249: “In addition to physicochemical protection, aridity-induced changes in substrate also exerted roles in regulating Q_{10_SOC} . Consistent with previous studies on moist soils^{48,49}, we observed that low substrate quality was associated with high Q_{10_SOC} , as indicated by the negative relationship of Q_{10_SOC} with D_{SOC} (**Fig. 4**), suggesting that the C quality-temperature hypothesis is also applicable in these water-limited ecosystems.”

Fig. 5 | Direct and indirect effects of climate, physical, chemical, substrate and microbial properties on the temperature sensitivity (Q_{10}) of SOC- and SIC-derived CO_2 in the topsoil (0–10 cm). a–b, Structural equation modeling was conducted for the Q_{10} of SOC-derived CO_2 (a) and SIC-derived CO_2 (b). The temperature sensitivity was estimated under field moisture conditions. Black dotted and solid arrows indicate negative and positive relationships, respectively, and gray arrows indicate insignificant relationships; the arrow width is proportional to the strength of the relationship, with the numbers adjacent to the arrows representing the standardized path coefficients. The multiple-layer rectangles represent the first component from the PCA conducted for the climate, physical, chemical, substrate and microbial properties. **c–d**, The standardized total effects (direct plus indirect effects) of different factors on Q_{10} of SOC-derived CO_2 (c) and SIC-derived CO_2 (d) derived from the SEM. MAT: mean annual temperature; AI: aridity index; OC-Ca and OC-Fe, the contents of SOC associated with Ca oxides and Fe bridges, respectively; OC-POM and OC-MAOM, the content of SOC stored

in the POM and MAOM fraction, respectively; CEC: cation exchange capacity; CAI, carbon availability index; D_{SOC} , SOC decomposability; B_div: bacterial diversity; F_div: fungal diversity.

[Comment 22] L186: The authors might consider removing this metric. It is now well understood that humus is an operationally defined metric, rather than something that is biologically relevant.

[Response] Thanks for the comment. Based on the comments here and above (*[Comment 21]*), we have removed the metric of humification index in the revised manuscript. Instead, we have added measurements and/or calculations of new metrics of substrate, including carbon availability index (please see the page 23, lines 403-407) and SOC decomposability (please see the page 23, lines 407-409) that could directly represent the biologically relevant substrate status in the revised manuscript.

[Comment 23] L226: Cation bridging mechanisms should be more thoroughly discussed.

[Response] Thanks for the comment. We have added some discussions on the cation bridging mechanisms in the revised manuscript.

Please see the page 15, lines 238-243: “Specifically, MAOM fractions can restrict oxygen diffusion and lead to compartmentalization of organic C substrates from enzymes, and these processes can be enhanced by Ca bridges and/or Fe oxides³⁴. In addition, Ca bridges and/or Fe oxides can also constrain substrate availability by forming both inner- and outer-sphere cation bridging between the negatively charged phyllosilicates and SOC⁴⁷. Either of these processes may suppress the temperature response of SOC decomposition^{27,30}.”

[Comment 24] L239: As suggested earlier, the authors might rely less on this interpretation.

[Response] Thanks so much for the comment. Based on the comments here and above (*[Comment 1]* and *[Comment 19]*), we have removed the use of metric oligotroph:copiotroph ratio in the revised manuscript.

[Comment 25] L248: This sentence is quite confusing the concepts need separating and clarifying.

[Response] Thanks for the comment. We have made it clear in the revised manuscript.

Please see the pages 15-16, lines 252-255: “This is because a higher pH and/or base cation (e.g., Ca^{2+} and Mg^{2+}) concentration may enhance the reverse reactions represented in Equations (1)

and (2) toward the absorption of CO₂ into the soil solution. Given that a low CO₂ emission rate might be more sensitive to temperature changes, high pH and/or base cations can thus enhance Q_{10_SIC} .”

[Comment 26] L258: *Not necessarily synonymous with warming, clarify - do the authors mean to compare SOC and SIC temperature or moisture sensitivity?*

[Response] Thanks for the comment. We have made it clear in the revised manuscript.

Please see the page 16, lines 264-266: “In conclusion, our findings provide evidence for opposing moisture effects on the temperature response of SOC and SIC in drylands, suggesting that drying will further enhance the temperature response of SIC but weaken that of SOC (Fig. 6).”

[Comment 27] L263: *Please expand on this if including, this is a little oversimplified*

<https://www.sciencedirect.com/science/article/pii/S0048969720331302>

<https://besjournals.onlinelibrary.wiley.com/doi/full/10.1111/1365-2745.13896>.

[Response] Thanks for the comment. We have expanded the potential effects of to nitrogen deposition on dryland soil carbon by citing these two recent studies in the revised manuscript.

Please see the page 16, lines 271-274: “Although some studies have shown that soil C in drylands is resistant to nitrogen deposition^{58,59}, a recent study concluded that global N fertilization results in releases of 7.5×10^{12} g C year⁻¹ from carbonates³⁹, and this value may be even underestimated⁸.”

[Comment 28] L274: *This is not a particularly useful figure in my opinion, "temperature sensitivity of SIC dissolution" IS SIC temperature sensitivity.*

[Response] Thanks so much for the comment. Based on the comments here and above (**[Comment 2]**), we have largely improved the graphical abstract (Figure 6) to demonstrate the directionality of the controls in more detail in the revised manuscript (please see the new Figure 6). Moreover, we have changed “temperature sensitivity of SIC dissolution” to “SIC temperature sensitivity” in the figure caption.

[Comment 29] L421: *Please acknowledge the water source/chemistry - this is especially important in carbonate soils as some water sources can agress carbonates.*

[Response] Thanks so much for the comment. Deionized water with pH about 7.3 was added. We have added these descriptions in the revised manuscript (please see the page 19, line 322).

[Comment 30] L460: As mentioned previously, the reasoning for these measurements has not been elucidated.

[Response] Thanks so much for the comment. Based on the comments here and above (*[Comment 12]* and *[Comment 23]*), we have added some descriptions (please see the page 4, lines 69-73) and discussions (please see the page 15, lines 238-243) on the effects of Ca bridges and/or Fe oxides on the temperature sensitivity of soil carbon decomposition in the revised manuscript.

[Comment 31] L480: Could the authors explain why SIC was not measured directly. This method of subtraction is usually accepted for determining SOC (although for the same reason could be considered erroneous). In practice it is more accurate to determine SIC using a manometer method and building a calibration curve. The method used by the authors may be biased in soils with negligible carbonate contents. Stephen J Hall (Iowa State University) has done some excellent but I believe still unpublished work on this.

[Response] Thanks so much for the comment. To be more accurate, we used the direct method (pressure calcimeter method) to estimate SIC content in the revised manuscript.

Please see the pages 22-23, lines 398-403: “SIC was determined by a pressure calcimeter method⁷¹. Briefly, 0.5 g soil was mixed with 2 mL 6 M HCl and reacted in a closed reaction vessel. Two hours later, the pressure was determined using the pressure transducer and voltage meter, and then the carbonate concentration was calculated using a calibration curve, which was obtained in the same way using known quantities of CaCO₃. The SIC content was finally determined by multiplying by a coefficient of 0.12, which is the mass proportion of C in calcium carbonate (0.12 g C g⁻¹ CaCO₃).”

Moreover, we have incorporated the new data of SIC into the mechanisms exploration of Q_{10_SOC} and Q_{10_SIC} changes along the aridity gradient in the revised manuscript (for example, please see the updated Fig. 4 and Fig. 5).

Fig. 4 | Correlations of the temperature sensitivity (Q_{10}) of SOC- and SIC-derived CO_2 emissions with factors related to climate, physical, chemical, substrate and microbial properties. The temperature sensitivity was estimated under field moisture conditions. Q_{10_SOC} and Q_{10_SIC} : Q_{10} of SOC-derived and SIC-derived CO_2 emissions, respectively; MAT: mean annual temperature; AI: aridity index; OC-Ca and OC-Fe, the contents of SOC associated with Ca oxides and Fe bridges, respectively; OC-POM and OC-MAOM, the content of SOC stored in the POM and MAOM fraction, respectively; CEC: cation exchange capacity; CAI, carbon availability index; D_{SOC} , SOC decomposability; B_{div}: bacterial diversity; F_{div}: fungal diversity. * $P < 0.05$; ** $P < 0.01$; *** $P < 0.001$.

[Comment 32] L483: Can the authors address the potential issues with this method as these determine pools that are operationally defined and not necessarily biologically relevant.

[Response] Thanks so much for the comment. Based on the comment, we have also calculated the decomposability of SOC (D_{SOC} , SOC decomposition rate per unit SOC content), which can represent the biological quality of SOC (Craine *et al.*, 2010, *Nature Geoscience*). We found that the recalcitrant/labile SOC ratio (R:L SOC) was significantly correlated with D_{SOC} (**Fig. R1**), suggesting that R:L SOC can represent biologically relevant quality of SOC to some extent. To better represent biologically relevant substrate quality, we have used D_{SOC} in the revised manuscript (please see the page 23, lines 407-409).

Fig. R1 | Relationships of the recalcitrant/labile SOC ratio (R:L SOC) with SOC decomposability (D_{soc}) along an aridity gradient. D_{soc} was estimated under moisture condition of 40% water holding capacity at 20°C.

[Comment 33] L523: Can the authors explain why a constrained ordination (like RDA) was not used for this purpose- if the intent was to reduce collinearity?

[Response] Thanks so much for the comment. The PCA has been widely used for dimensionality reduction for a target group of factors (e.g., climate, physical, chemical, substrate and microbial property in this study) (for example, Chen *et al.*, 2019, *Nature Communications*; Qin *et al.*, 2021, *Science Advances*). Because PCA is a non-constrained ordination, which is conducted to simplify information on data and to represent the individual variables included in this group. The RDA is a constrained ordination, which can obtain a new variable for the individual variables included in this group under the constraint of various other factors. Based on these fundamental principles, the relationships shown in the SEM are easier to interpret when using PCA for the dimensionality reduction as PCA is conducted to represent the data information of the individual factors; however, the relationships shown in the SEM are not easy to interpret if using RDA for the dimensionality reduction as RDA also considers the effects of various other factors.

Overall, we appreciate the three reviewers' insightful comments. These comments enabled us to have much deeper exploration of the data, to explore the mechanisms underling the different responses of Q_{10_SOC} and Q_{10_SOC} to moisture changes, to discussion the limitations of our study,

and to explore the implications of our findings. By addressing these comments, we feel that the revised manuscript has been greatly improved. Thank you!

REVIEWER COMMENTS

Reviewer #2 (Remarks to the Author):

I am happy to see the revised version of the paper. All of my comments were carefully addressed (as well as for the other two reviewers).

I am satisfied with the changes that have been made. In my view the paper is much improved and is ready for publication.

Congrats!

Reviewer #3 (Remarks to the Author):

Overall, I am impressed with the attention to detail that has gone into addressing the comments made by all reviewers. I agree with the authors, that the manuscript is significantly improved.

However, the major concern I still have with the presentation and interpretation of these data is the lack of attention given to microbial mechanisms, given that the authors have chosen to measure and include these metrics. This has still not been adequately addressed. From my point of view, the authors have at least two options:

1) Revisit the interpretation of SOC and SIC decomposition and their interacting mechanisms from a microbially-focused standpoint to the extent that the data will allow and incorporate this into the manuscript at all levels. Unfortunately, looking at the data, and through my interpretation of the new figures (much appreciated) there may not be a lot there to examine. Which leads us to option 2.

2) Remove the microbial community data altogether and save it for another paper where these mechanisms can be given the attention they deserve, once further data has been collected. If this were my paper, this would be my decision. I am aware that there is little literature and data on microbial community structure/functional relationships in arid-calcareous soils, and therefore not a lot to draw from. This is not a good reason to keep these data in, given they don't contribute significantly to the already scant literature. To me the microbial data is still just 'hanging' off the back of what is already an excellent paper, that frankly doesn't need it to remain excellent.

Even the fact that none of the hypotheses mention microbial communities or microbial function, indicates that this is not a well-integrated piece of the data.

I hope that this suggestion goes towards improving what is a fantastic piece of work.

Response to reviewers' comments on

“Drought exacerbates dryland soil carbon loss from inorganic carbon under warming”

Response to Reviewer #2

Comments to the Author:

[Comment 1] I am happy to see the revised version of the paper. All of my comments were carefully addressed (as well as for the other two reviewers).

I am satisfied with the changes that have been made. In my view the paper is much improved and is ready for publication.

Congrats!

[Response] Thanks for the reviewer's positive assessments and all the efforts in reviewing our manuscript. Thanks!

Response to Reviewer #3

Comments to the Author:

[Comment 1] Overall, I am impressed with the attention to detail that has gone into addressing the comments made by all reviewers. I agree with the authors, that the manuscript is significantly improved.

However, the major concern I still have with the presentation and interpretation of these data is the lack of attention given to microbial mechanisms, given that the authors have chosen to measure and include these metrics. This has still not been adequately addressed. From my point of view, the authors have at least two options:

1) Revisit the interpretation of SOC and SIC decomposition and their interacting mechanisms from a microbially-focused standpoint to the extent that the data will allow and incorporate this into the manuscript at all levels. Unfortunately, looking at the data, and through my interpretation of the new figures (much appreciated) there may not be a lot there to examine. Which leads us to option 2.

2) Remove the microbial community data altogether and save it for another paper where these mechanisms can be given the attention they deserve, once further data has been collected. If this were my paper, this would be my decision. I am aware that there is little literature and data on microbial community structure/functional relationships in arid-calcareous soils, and therefore not a lot to draw from. This is not a good reason to keep these data in, given they don't contribute significantly to the already scant literature. To me the microbial data is still just 'hanging' off the back of what is already an excellent paper, that frankly doesn't need it to remain excellent.

Even the fact that none of the hypotheses mention microbial communities or microbial function, indicates that this is not a well-integrated piece of the data.

I hope that this suggestion goes towards improving what is a fantastic piece of work.

[Response] Thanks for the reviewer's positive assessments and all insightful comments. Based on the analyses, we found that the microbial communities indeed played very small roles in explaining the spatial variations in Q_{10} of SOC or SIC along our studies aridity gradient. We agree with the reviewer that incorporating the microbial community data did not improve the manuscript much. We thus adopted the second option provided by the reviewer, and we have removed the microbial community data in the revised manuscript. Thanks!